# Non-convex Statistical Optimization for Sparse Tensor Graphical Model

**Wei Sun**
Yahoo Labs
Sunnyvale, CA
sunweisurrey@yahoo-inc.com

**Zhaoran Wang**
Department of Operations Research
and Financial Engineering
Princeton University
Princeton, NJ
zhaoran@princeton.edu

**Han Liu**
Department of Operations Research
and Financial Engineering
Princeton University
Princeton, NJ
hanliu@princeton.edu

**Guang Cheng**
Department of Statistics
Purdue University
West Lafayette, IN
chengg@stat.purdue.edu

## Abstract

We consider the estimation of sparse graphical models that characterize the dependency structure of high-dimensional tensor-valued data. To facilitate the estimation of the precision matrix corresponding to each way of the tensor, we assume the data follow a tensor normal distribution whose covariance has a Kronecker product structure. The penalized maximum likelihood estimation of this model involves minimizing a non-convex objective function. In spite of the non-convexity of this estimation problem, we prove that an alternating minimization algorithm, which iteratively estimates each sparse precision matrix while fixing the others, attains an estimator with the optimal statistical rate of convergence as well as consistent graph recovery. Notably, such an estimator achieves estimation consistency with only one tensor sample, which is unobserved in previous work. Our theoretical results are backed by thorough numerical studies.

## 1 Introduction

High-dimensional tensor-valued data are prevalent in many fields such as personalized recommendation systems and brain imaging research [1, 2]. Traditional recommendation systems are mainly based on the user-item matrix, whose entry denotes each user's preference for a particular item. To incorporate additional information into the analysis, such as the temporal behavior of users, we need to consider a user-item-time tensor. For another example, functional magnetic resonance imaging (fMRI) data can be viewed as a three way (third-order) tensor since it contains the brain measurements taken on different locations over time for various experimental conditions. Also, in the example of microarray study for aging [3], thousands of gene expression measurements are recorded on 16 tissue types on 40 mice with varying ages, which forms a four way gene-tissue-mouse-age tensor.

In this paper, we study the estimation of conditional independence structure within tensor data. For example, in the microarray study for aging we are interested in the dependency structure across different genes, tissues, ages and even mice. Assuming data are drawn from a tensor normal distribution, a straightforward way to estimate this structure is to vectorize the tensor and estimate the underlying Gaussian graphical model associated with the vector. Such an approach ignores the tensor structure

and requires estimating a rather high dimensional precision matrix with insufficient sample size. For instance, in the aforementioned fMRI application the sample size is one if we aim to estimate the dependency structure across different locations, time and experimental conditions. To address such a problem, a popular approach is to assume the covariance matrix of the tensor normal distribution is separable in the sense that it is the Kronecker product of small covariance matrices, each of which corresponds to one way of the tensor. Under this assumption, our goal is to estimate the precision matrix corresponding to each way of the tensor. See §1.1 for a detailed survey of previous work.

Despite the fact that the assumption of the Kronecker product structure of covariance makes the statistical model much more parsimonious, it poses significant challenges. In particular, the penalized negative log-likelihood function is non-convex with respect to the unknown sparse precision matrices. Consequently, there exists a gap between computational and statistical theory. More specifically, as we will show in §1.1, existing literature mostly focuses on establishing the existence of a local optimum that has desired statistical guarantees, rather than offering efficient algorithmic procedures that provably achieve the desired local optima. In contrast, we analyze an alternating minimization algorithm which iteratively minimizes the non-convex objective function with respect to each individual precision matrix while fixing the others. The established theoretical guarantees of the proposed algorithm are as follows. Suppose that we have $n$ observations from a $K$-th order tensor normal distribution. We denote by $m_k$, $s_k$, $d_k$ ($k = 1, \ldots, K$) the dimension, sparsity, and max number of non-zero entries in each row of the precision matrix corresponding to the $k$-th way of the tensor. Besides, we define $m = \prod_{k=1}^{K} m_k$. The $k$-th precision matrix estimator from our alternating minimization algorithm achieves a $\sqrt{m_k(m_k + s_k)\log m_k/(nm)}$ statistical rate of convergence in Frobenius norm, which is minimax-optimal since this is the best rate one can obtain even when the rest $K - 1$ true precision matrices are known [4]. Furthermore, under an extra irrepresentability condition, we establish a $\sqrt{m_k \log m_k/(nm)}$ rate of convergence in max norm, which is also optimal, and a $d_k\sqrt{m_k \log m_k/(nm)}$ rate of convergence in spectral norm. These estimation consistency results and a sufficiently large signal strength condition further imply the model selection consistency of recovering all the edges. A notable implication of these results is that, when $K \geq 3$, our alternating minimization algorithm can achieve estimation consistency in Frobenius norm even if we only have access to one tensor sample, which is often the case in practice. This phenomenon is unobserved in previous work. Finally, we conduct extensive experiments to evaluate the numerical performance of the proposed alternating minimization method. Under the guidance of theory, we propose a way to significantly accelerate the algorithm without sacrificing the statistical accuracy.

## 1.1 Related work and our contribution

A special case of our sparse tensor graphical model when $K = 2$ is the sparse matrix graphical model, which is studied by [5–8]. In particular, [5] and [6] only establish the existence of a local optima with desired statistical guarantees. Meanwhile, [7] considers an algorithm that is similar to ours. However, the statistical rates of convergence obtained by [6, 7] are much slower than ours when $K = 2$. See Remark 3.6 in §3.1 for a detailed comparison. For $K = 2$, our statistical rate of convergence in Frobenius norm recovers the result of [5]. In other words, our theory confirms that the desired local optimum studied by [5] not only exists, but is also attainable by an efficient algorithm. In addition, for matrix graphical model, [8] establishes the statistical rates of convergence in spectral and Frobenius norms for the estimator attained by a similar algorithm. Their results achieve estimation consistency in spectral norm with only one matrix observation. However, their rate is slower than ours with $K = 2$. See Remark 3.11 in §3.2 for a detailed discussion. Furthermore, we allow $K$ to increase and establish estimation consistency even in Frobenius norm for $n = 1$. Most importantly, all these results focus on matrix graphical model and can not handle the aforementioned motivating applications such as the gene-tissue-mouse-age tensor dataset.

In the context of sparse tensor graphical model with a general $K$, [9] shows the existence of a local optimum with desired rates, but does not prove whether there exists an efficient algorithm that provably attains such a local optimum. In contrast, we prove that our alternating minimization algorithm achieves an estimator with desired statistical rates. To achieve it, we apply a novel theoretical framework to separately consider the population and sample optimizers, and then establish the one-step convergence for the population optimizer (Theorem 3.1) and the optimal rate of convergence for the sample optimizer (Theorem 3.4). A new concentration result (Lemma B.1) is developed for this purpose, which is also of independent interest. Moreover, we establish additional theoretical

guarantees including the optimal rate of convergence in max norm, the estimation consistency in spectral norm, and the graph recovery consistency of the proposed sparse precision matrix estimator.

In addition to the literature on graphical models, our work is also closely related to a recent line of research on alternating minimization for non-convex optimization problems [10–13]. These existing results mostly focus on problems such as dictionary learning, phase retrieval and matrix decomposition. Hence, our statistical model and analysis are completely different from theirs. Also, our paper is related to a recent line of work on tensor decomposition. See, e.g., [14–17] and the references therein. Compared with them, our work focuses on the graphical model structure within tensor-valued data.

**Notation:** For a matrix $\mathbf{A} = (\mathbf{A}_{i,j}) \in \mathbb{R}^{d \times d}$, we denote $\|\mathbf{A}\|_\infty, \|\mathbf{A}\|_2, \|\mathbf{A}\|_F$ as its max, spectral, and Frobenius norm, respectively. We define $\|\mathbf{A}\|_{1,\mathrm{off}} := \sum_{i \neq j} |\mathbf{A}_{i,j}|$ as its off-diagonal $\ell_1$ norm and $\|\mathbf{A}\|_\infty := \max_i \sum_j |\mathbf{A}_{i,j}|$ as the maximum absolute row sum. Denote $\mathrm{vec}(\mathbf{A})$ as the vectorization of $\mathbf{A}$ which stacks the columns of $\mathbf{A}$. Let $\mathrm{tr}(\mathbf{A})$ be the trace of $\mathbf{A}$. For an index set $\mathbb{S} = \{(i,j), i, j \in \{1, \ldots, d\}\}$, we define $[\mathbf{A}]_{\mathbb{S}}$ as the matrix whose entry indexed by $(i,j) \in \mathbb{S}$ is equal to $\mathbf{A}_{i,j}$, and zero otherwise. We denote $\mathbb{1}_d$ as the identity matrix with dimension $d \times d$. Throughout this paper, we use $C, C_1, C_2, \ldots$ to denote generic absolute constants, whose values may vary from line to line.

## 2 Sparse tensor graphical model

### 2.1 Preliminary

We employ the tensor notations used by [18]. Throughout this paper, higher order tensors are denoted by boldface Euler script letters, e.g. $\mathcal{T}$. We consider a $K$-th order tensor $\mathcal{T} \in \mathbb{R}^{m_1 \times m_2 \times \cdots \times m_K}$. When $K = 1$ it reduces to a vector and when $K = 2$ it reduces to a matrix. The $(i_1, \ldots, i_K)$-th element of the tensor $\mathcal{T}$ is denoted to be $\mathcal{T}_{i_1, \ldots, i_K}$. Meanwhile, we define the vectorization of $\mathcal{T}$ as $\mathrm{vec}(\mathcal{T}) := (\mathcal{T}_{1,1,\ldots,1}, \ldots, \mathcal{T}_{m_1,1,\ldots,1}, \ldots, \mathcal{T}_{1,m_2,\ldots,m_K}, \mathcal{T}_{m_1,m_2,\ldots,m_K})^\top \in \mathbb{R}^m$ with $m = \prod_k m_k$. In addition, we define the Frobenius norm of a tensor $\mathcal{T}$ as $\|\mathcal{T}\|_F := \left(\sum_{i_1,\ldots,i_K} \mathcal{T}_{i_1,\ldots,i_K}^2\right)^{1/2}$.

For tensors, a fiber refers to the higher order analogue of the row and column of matrices. A fiber is obtained by fixing all but one of the indices of the tensor, e.g., the mode-$k$ fiber of $\mathcal{T}_{(k)}$ is given by $\mathcal{T}_{i_1,\ldots,i_{k-1},:,i_{k+1},\ldots,i_K}$. Matricization, also known as unfolding, is the process to transform a tensor into a matrix. We denote $\mathcal{T}_{(k)}$ as the mode-$k$ matricization of a tensor $\mathcal{T}$, which arranges the mode-$k$ fibers to be the columns of the resulting matrix. Another useful operation in tensors is the $k$-mode product. The $k$-mode product of a tensor $\mathcal{T} \in \mathbb{R}^{m_1 \times m_2 \times \cdots \times m_K}$ with a matrix $\mathbf{A} \in \mathbb{R}^{J \times m_k}$ is denoted as $\mathcal{T} \times_k \mathbf{A}$ and is of the size $m_1 \times \cdots \times m_{k-1} \times J \times m_{k+1} \times \cdots \times m_K$. Its entry is defined as $(\mathcal{T} \times_k \mathbf{A})_{i_1,\ldots,i_{k-1},j,i_{k+1},\ldots,i_K} := \sum_{i_k=1}^{m_k} \mathcal{T}_{i_1,\ldots,i_K} \mathbf{A}_{j,i_k}$. In addition, for a list of matrices $\{\mathbf{A}_1, \ldots, \mathbf{A}_K\}$ with $\mathbf{A}_k \in \mathbb{R}^{m_k \times m_k}$, $k = 1, \ldots, K$, we define $\mathcal{T} \times \{\mathbf{A}_1, \ldots, \mathbf{A}_K\} := \mathcal{T} \times_1 \mathbf{A}_1 \times_2 \cdots \times_K \mathbf{A}_K$.

### 2.2 Model

A tensor $\mathcal{T} \in \mathbb{R}^{m_1 \times m_2 \times \cdots \times m_K}$ follows the tensor normal distribution with zero mean and covariance matrices $\mathbf{\Sigma}_1, \ldots, \mathbf{\Sigma}_K$, denoted as $\mathcal{T} \sim \mathrm{TN}(\mathbf{0}; \mathbf{\Sigma}_1, \ldots, \mathbf{\Sigma}_K)$, if its probability density function is

$$p(\mathcal{T}|\mathbf{\Sigma}_1, \ldots, \mathbf{\Sigma}_K) = (2\pi)^{-m/2} \left\{\prod_{k=1}^{K} |\mathbf{\Sigma}_k|^{-m/(2m_k)}\right\} \exp\left(-\|\mathcal{T} \times \mathbf{\Sigma}^{-1/2}\|_F^2/2\right), \qquad (2.1)$$

where $m = \prod_{k=1}^{K} m_k$ and $\mathbf{\Sigma}^{-1/2} := \{\mathbf{\Sigma}_1^{-1/2}, \ldots, \mathbf{\Sigma}_K^{-1/2}\}$. When $K = 1$, this tensor normal distribution reduces to the vector normal distribution with zero mean and covariance $\mathbf{\Sigma}_1$. According to [9, 18], it can be shown that $\mathcal{T} \sim \mathrm{TN}(\mathbf{0}; \mathbf{\Sigma}_1, \ldots, \mathbf{\Sigma}_K)$ if and only if $\mathrm{vec}(\mathcal{T}) \sim \mathrm{N}(\mathrm{vec}(\mathbf{0}); \mathbf{\Sigma}_K \otimes \cdots \otimes \mathbf{\Sigma}_1)$, where $\mathrm{vec}(\mathbf{0}) \in \mathbb{R}^m$ and $\otimes$ is the matrix Kronecker product.

We consider the parameter estimation for the tensor normal model. Assume that we observe independently and identically distributed tensor samples $\mathcal{T}_1, \ldots, \mathcal{T}_n$ from $\mathrm{TN}(\mathbf{0}; \mathbf{\Sigma}_1^*, \ldots, \mathbf{\Sigma}_K^*)$. We aim to estimate the true covariance matrices $(\mathbf{\Sigma}_1^*, \ldots, \mathbf{\Sigma}_K^*)$ and their corresponding true precision matrices $(\mathbf{\Omega}_1^*, \ldots, \mathbf{\Omega}_K^*)$ where $\mathbf{\Omega}_k^* = \mathbf{\Sigma}_k^{*-1}$ $(k = 1, \ldots, K)$. To address the identifiability issue in the parameterization of the tensor normal distribution, we assume that $\|\mathbf{\Omega}_k^*\|_F = 1$ for $k = 1, \ldots, K$. This renormalization assumption does not change the graph structure of the original precision matrix.

A standard approach to estimate $\mathbf{\Omega}_k^*$, $k = 1, \ldots, K$, is to use the maximum likelihood method via (2.1). Up to a constant, the negative log-likelihood function of the tensor normal distribution is $\mathrm{tr}[\mathbf{S}(\mathbf{\Omega}_K \otimes \cdots \otimes \mathbf{\Omega}_1)] - \sum_{k=1}^K (m/m_k) \log |\mathbf{\Omega}_k|$, where $\mathbf{S} := \frac{1}{n} \sum_{i=1}^n \mathrm{vec}(\mathcal{T}_i) \mathrm{vec}(\mathcal{T}_i)^\top$. To encourage the sparsity of each precision matrix in the high-dimensional scenario, we consider a penalized log-likelihood estimator, which is obtained by minimizing

$$q_n(\mathbf{\Omega}_1, \ldots, \mathbf{\Omega}_K) := \frac{1}{m} \mathrm{tr}[\mathbf{S}(\mathbf{\Omega}_K \otimes \cdots \otimes \mathbf{\Omega}_1)] - \sum_{k=1}^K \frac{1}{m_k} \log |\mathbf{\Omega}_k| + \sum_{k=1}^K P_{\lambda_k}(\mathbf{\Omega}_k), \qquad (2.2)$$

where $P_{\lambda_k}(\cdot)$ is a penalty function indexed by the tuning parameter $\lambda_k$. In this paper, we focus on the lasso penalty [19], i.e., $P_{\lambda_k}(\mathbf{\Omega}_k) = \lambda_k \|\mathbf{\Omega}_k\|_{1,\mathrm{off}}$. This estimation procedure applies similarly to a broad family of other penalty functions.

We name the penalized model from (2.2) as the sparse tensor graphical model. It reduces to the sparse vector graphical model [20, 21] when $K = 1$, and the sparse matrix graphical model [5–8] when $K = 2$. Our framework generalizes them to fulfill the demand of capturing the graphical structure of higher order tensor-valued data.

## 2.3 Estimation

This section introduces the estimation procedure for the sparse tensor graphical model. A computationally efficient algorithm is provided to estimate the precision matrix for each way of the tensor.

Recall that in (2.2), $q_n(\mathbf{\Omega}_1, \ldots, \mathbf{\Omega}_K)$ is jointly non-convex with respect to $\mathbf{\Omega}_1, \ldots, \mathbf{\Omega}_K$. Nevertheless, $q_n(\mathbf{\Omega}_1, \ldots, \mathbf{\Omega}_K)$ is a bi-convex problem since $q_n(\mathbf{\Omega}_1, \ldots, \mathbf{\Omega}_K)$ is convex in $\mathbf{\Omega}_k$ when the rest $K - 1$ precision matrices are fixed. The bi-convex property plays a critical role in our algorithm construction and its theoretical analysis in §3.

According to its bi-convex property, we propose to solve this non-convex problem by alternatively update one precision matrix with other matrices fixed. Note that, for any $k = 1, \ldots, K$, minimizing (2.2) with respect to $\mathbf{\Omega}_k$ while fixing the rest $K - 1$ precision matrices is equivalent to minimizing

$$L(\mathbf{\Omega}_k) := \frac{1}{m_k} \mathrm{tr}(\mathbf{S}_k \mathbf{\Omega}_k) - \frac{1}{m_k} \log |\mathbf{\Omega}_k| + \lambda_k \|\mathbf{\Omega}_k\|_{1,\mathrm{off}}. \qquad (2.3)$$

Here $\mathbf{S}_k := \frac{m_k}{nm} \sum_{i=1}^n \mathbf{V}_i^k \mathbf{V}_i^{k\top}$, where $\mathbf{V}_i^k := \left[ \mathcal{T}_i \times \left\{ \mathbf{\Omega}_1^{1/2}, \ldots, \mathbf{\Omega}_{k-1}^{1/2}, \mathbb{1}_{m_k}, \mathbf{\Omega}_{k+1}^{1/2}, \ldots, \mathbf{\Omega}_K^{1/2} \right\} \right]_{(k)}$ with $\times$ the tensor product operation and $[\cdot]_{(k)}$ the mode-$k$ matricization operation defined in §2.1. The result in (2.3) can be shown by noting that $\mathbf{V}_i^k = [\mathcal{T}_i]_{(k)} \left( \mathbf{\Omega}_K^{1/2} \otimes \cdots \otimes \mathbf{\Omega}_{k+1}^{1/2} \otimes \mathbf{\Omega}_{k-1}^{1/2} \otimes \cdots \otimes \mathbf{\Omega}_1^{1/2} \right)^\top$ according to the properties of mode-$k$ matricization shown by [18]. Hereafter, we drop the superscript $k$ of $\mathbf{V}_i^k$ if there is no confusion. Note that minimizing (2.3) corresponds to estimating vector-valued Gaussian graphical model and can be solved efficiently via the glasso algorithm [21].

---

**Algorithm 1** Solve sparse tensor graphical model via Tensor lasso (Tlasso)

---

1: **Input:** Tensor samples $\mathcal{T}_1 \ldots, \mathcal{T}_n$, tuning parameters $\lambda_1, \ldots, \lambda_K$, max number of iterations $T$.
2: **Initialize** $\mathbf{\Omega}_1^{(0)}, \ldots, \mathbf{\Omega}_K^{(0)}$ randomly as symmetric and positive definite matrices and set $t = 0$.
3: **Repeat**:
4: $\quad t = t + 1$.
5: $\quad$ **For** $k = 1, \ldots, K$:
6: $\quad\quad$ Given $\mathbf{\Omega}_1^{(t)}, \ldots, \mathbf{\Omega}_{k-1}^{(t)}, \mathbf{\Omega}_{k+1}^{(t-1)}, \ldots, \mathbf{\Omega}_K^{(t-1)}$, solve (2.3) for $\mathbf{\Omega}_k^{(t)}$ via glasso [21].
7: $\quad\quad$ Normalize $\mathbf{\Omega}_k^{(t)}$ such that $\|\mathbf{\Omega}_k^{(t)}\|_F = 1$.
8: **End For**
9: **Until** $t = T$.
10: **Output:** $\widehat{\mathbf{\Omega}}_k = \mathbf{\Omega}_k^{(T)}$ $(k = 1, \ldots, K)$.

---

The details of our Tensor lasso (Tlasso) algorithm are shown in Algorithm 1. It starts with a random initialization and then alternatively updates each precision matrix until it converges. In §3, we will illustrate that the statistical properties of the obtained estimator are insensitive to the choice of the initialization (see the discussion following Theorem 3.5).

# 3 Theory of statistical optimization

We first prove the estimation errors in Frobenius norm, max norm, and spectral norm, and then provide the model selection consistency of our Tlasso estimator. We defer all the proofs to the appendix.

## 3.1 Estimation error in Frobenius norm

Based on the penalized log-likelihood in (2.2), we define the population log-likelihood function as

$$q(\mathbf{\Omega}_1, \ldots, \mathbf{\Omega}_K) := \frac{1}{m} \mathbb{E}\left\{ \text{tr}\left[ \text{vec}(\mathcal{T})\text{vec}(\mathcal{T})^\top (\mathbf{\Omega}_K \otimes \cdots \otimes \mathbf{\Omega}_1) \right] \right\} - \sum_{k=1}^{K} \frac{1}{m_k} \log |\mathbf{\Omega}_k|. \quad (3.1)$$

By minimizing $q(\mathbf{\Omega}_1, \ldots, \mathbf{\Omega}_K)$ with respect to $\mathbf{\Omega}_k$, $k = 1, \ldots, K$, we obtain the population minimization function with the parameter $\mathbf{\Omega}_{[K]-k} := \{\mathbf{\Omega}_1, \ldots, \mathbf{\Omega}_{k-1}, \mathbf{\Omega}_{k+1}, \ldots, \mathbf{\Omega}_K\}$, i.e.,

$$M_k(\mathbf{\Omega}_{[K]-k}) := \operatorname*{argmin}_{\mathbf{\Omega}_k} q(\mathbf{\Omega}_1, \ldots, \mathbf{\Omega}_K). \quad (3.2)$$

**Theorem 3.1.** For any $k = 1, \ldots, K$, if $\mathbf{\Omega}_j$ ($j \neq k$) satisfies $\text{tr}(\mathbf{\Sigma}_j^* \mathbf{\Omega}_j) \neq 0$, then the population minimization function in (3.2) satisfies $M_k(\mathbf{\Omega}_{[K]-k}) = m\left[ m_k \prod_{j \neq k} \text{tr}(\mathbf{\Sigma}_j^* \mathbf{\Omega}_j) \right]^{-1} \mathbf{\Omega}_k^*$.

Theorem 3.1 shows a surprising phenomenon that the population minimization function recovers the true precision matrix up to a constant in only one iteration. If $\mathbf{\Omega}_j = \mathbf{\Omega}_j^*$, $j \neq k$, then $M_k(\mathbf{\Omega}_{[K]-k}) = \mathbf{\Omega}_k^*$. Otherwise, after a normalization such that $\|M_k(\mathbf{\Omega}_{[K]-k})\|_F = 1$, the normalized population minimization function still fully recovers $\mathbf{\Omega}_k^*$. This observation suggests that setting $T = 1$ in Algorithm 1 is sufficient. Such a suggestion will be further supported by our numeric results.

In practice, when (3.1) is unknown, we can approximate it via its sample version $q_n(\mathbf{\Omega}_1, \ldots, \mathbf{\Omega}_K)$ defined in (2.2), which gives rise to the statistical error in the estimation procedure. Analogously to (3.2), we define the sample-based minimization function with parameter $\mathbf{\Omega}_{[K]-k}$ as

$$\widehat{M}_k(\mathbf{\Omega}_{[K]-k}) := \operatorname*{argmin}_{\mathbf{\Omega}_k} q_n(\mathbf{\Omega}_1, \ldots, \mathbf{\Omega}_K). \quad (3.3)$$

In order to prove the estimation error, it remains to quantify the statistical error induced from finite samples. The following two regularity conditions are assumed for this purpose.

**Condition 3.2** (Bounded Eigenvalues). For any $k = 1, \ldots, K$, there is a constant $C_1 > 0$ such that,

$$0 < C_1 \leq \lambda_{\min}(\mathbf{\Sigma}_k^*) \leq \lambda_{\max}(\mathbf{\Sigma}_k^*) \leq 1/C_1 < \infty,$$

where $\lambda_{\min}(\mathbf{\Sigma}_k^*)$ and $\lambda_{\max}(\mathbf{\Sigma}_k^*)$ refer to the minimal and maximal eigenvalue of $\mathbf{\Sigma}_k^*$, respectively.

Condition 3.2 requires the uniform boundedness of the eigenvalues of true covariance matrices $\mathbf{\Sigma}_k^*$. It has been commonly assumed in the graphical model literature [22].

**Condition 3.3** (Tuning). For any $k = 1, \ldots, K$ and some constant $C_2 > 0$, the tuning parameter $\lambda_k$ satisfies $1/C_2 \sqrt{\log m_k/(nmm_k)} \leq \lambda_k \leq C_2 \sqrt{\log m_k/(nmm_k)}$.

Condition 3.3 specifies the choice of the tuning parameters. In practice, a data-driven tuning procedure [23] can be performed to approximate the optimal choice of the tuning parameters.

Before characterizing the statistical error, we define a sparsity parameter for $\mathbf{\Omega}_k^*$, $k = 1, \ldots, K$. Let $\mathbb{S}_k := \{(i,j) : [\mathbf{\Omega}_k^*]_{i,j} \neq 0\}$. Denote the sparsity parameter $s_k := |\mathbb{S}_k| - m_k$, which is the number of nonzero entries in the off-diagonal component of $\mathbf{\Omega}_k^*$. For each $k = 1, \ldots, K$, we define $\mathbb{B}(\mathbf{\Omega}_k^*)$ as the set containing $\mathbf{\Omega}_k^*$ and its neighborhood for some sufficiently large constant radius $\alpha > 0$, i.e.,

$$\mathbb{B}(\mathbf{\Omega}_k^*) := \{\mathbf{\Omega} \in \mathbb{R}^{m_k \times m_k} : \mathbf{\Omega} = \mathbf{\Omega}^\top; \mathbf{\Omega} \succ 0; \|\mathbf{\Omega} - \mathbf{\Omega}_k^*\|_F \leq \alpha\}. \quad (3.4)$$

**Theorem 3.4.** Assume Conditions 3.2 and 3.3 hold. For any $k = 1, \ldots, K$, the statistical error of the sample-based minimization function defined in (3.3) satisfies that, for any fixed $\mathbf{\Omega}_j \in \mathbb{B}(\mathbf{\Omega}_j^*)$ ($j \neq k$),

$$\left\| \widehat{M}_k(\mathbf{\Omega}_{[K]-k}) - M_k(\mathbf{\Omega}_{[K]-k}) \right\|_F = O_P\left( \sqrt{\frac{m_k(m_k + s_k)\log m_k}{nm}} \right), \quad (3.5)$$

where $M_k(\mathbf{\Omega}_{[K]-k})$ and $\widehat{M}_k(\mathbf{\Omega}_{[K]-k})$ are defined in (3.2) and (3.3), and $m = \prod_{k=1}^{K} m_k$.

Theorem 3.4 establishes the statistical error associated with $\widehat{M}_k(\boldsymbol{\Omega}_{[K]-k})$ for arbitrary $\boldsymbol{\Omega}_j \in \mathbb{B}(\boldsymbol{\Omega}_j^*)$ with $j \neq k$. In comparison, previous work on the existence of a local solution with desired statistical property only establishes theorems similar to Theorem 3.4 for $\boldsymbol{\Omega}_j = \boldsymbol{\Omega}_j^*$ with $j \neq k$. The extension to an arbitrary $\boldsymbol{\Omega}_j \in \mathbb{B}(\boldsymbol{\Omega}_j^*)$ involves non-trivial technical barriers. Particularly, we first establish the rate of convergence of the difference between a sample-based quadratic form with its expectation (Lemma B.1) via concentration of Lipschitz functions of Gaussian random variables [24]. This result is also of independent interest. We then carefully characterize the rate of convergence of $\mathbf{S}_k$ defined in (2.3) (Lemma B.2). Finally, we develop (3.5) using the results for vector-valued graphical models developed by [25].

According to Theorem 3.1 and Theorem 3.4, we obtain the rate of convergence of the Tlasso estimator in terms of Frobenius norm, which is our main result.

**Theorem 3.5.** Assume that Conditions 3.2 and 3.3 hold. For any $k = 1, \ldots, K$, if the initialization satisfies $\boldsymbol{\Omega}_j^{(0)} \in \mathbb{B}(\boldsymbol{\Omega}_j^*)$ for any $j \neq k$, then the estimator $\widehat{\boldsymbol{\Omega}}_k$ from Algorithm 1 with $T = 1$ satisfies,

$$\left\| \widehat{\boldsymbol{\Omega}}_k - \boldsymbol{\Omega}_k^* \right\|_F = O_P\left( \sqrt{\frac{m_k(m_k + s_k)\log m_k}{nm}} \right), \tag{3.6}$$

where $m = \prod_{k=1}^{K} m_k$ and $\mathbb{B}(\boldsymbol{\Omega}_j^*)$ is defined in (3.4).

Theorem 3.5 suggests that as long as the initialization is within a constant distance to the truth, our Tlasso algorithm attains a consistent estimator after only one iteration. This initialization condition $\boldsymbol{\Omega}_j^{(0)} \in \mathbb{B}(\boldsymbol{\Omega}_j^*)$ trivially holds since for any $\boldsymbol{\Omega}_j^{(0)}$ that is positive definite and has unit Frobenius norm, we have $\|\boldsymbol{\Omega}_j^{(0)} - \boldsymbol{\Omega}_k^*\|_F \leq 2$ by noting that $\|\boldsymbol{\Omega}_k^*\|_F = 1$ $(k = 1, \ldots, K)$ for the identifiability of the tensor normal distribution. In literature, [9] shows that there exists a local minimizer of (2.2) whose convergence rate can achieve (3.6). However, it is unknown if their algorithm can find such minimizer since there could be many other local minimizers.

A notable implication of Theorem 3.5 is that, when $K \geq 3$, the estimator from our Tlasso algorithm can achieve estimation consistency even if we only have access to one observation, i.e., $n = 1$, which is often the case in practice. To see it, suppose that $K = 3$ and $n = 1$. When the dimensions $m_1, m_2$, and $m_3$ are of the same order of magnitude and $s_k = O(m_k)$ for $k = 1, 2, 3$, all the three error rates corresponding to $k = 1, 2, 3$ in (3.6) converge to zero.

This result indicates that the estimation of the $k$-th precision matrix takes advantage of the information from the $j$-th way $(j \neq k)$ of the tensor data. Consider a simple case that $K = 2$ and one precision matrix $\boldsymbol{\Omega}_1^* = \mathbb{1}_{m_1}$ is known. In this scenario the rows of the matrix data are independent and hence the effective sample size for estimating $\boldsymbol{\Omega}_2^*$ is in fact $nm_1$. The optimality result for the vector-valued graphical model [4] implies that the optimal rate for estimating $\boldsymbol{\Omega}_2^*$ is $\sqrt{(m_2 + s_2)\log m_2/(nm_1)}$, which matches our result in (3.6). Therefore, the rate in (3.6) obtained by our Tlasso estimator is minimax-optimal since it is the best rate one can obtain even when $\boldsymbol{\Omega}_j^*$ $(j \neq k)$ are known. As far as we know, this phenomenon has not been discovered by any previous work in tensor graphical model.

**Remark 3.6.** For $K = 2$, our tensor graphical model reduces to matrix graphical model with Kronecker product covariance structure [5–8]. In this case, the rate of convergence of $\widehat{\boldsymbol{\Omega}}_1$ in (3.6) reduces to $\sqrt{(m_1 + s_1)\log m_1/(nm_2)}$, which is much faster than $\sqrt{m_2(m_1 + s_1)(\log m_1 + \log m_2)/n}$ established by [6] and $\sqrt{(m_1 + m_2)\log[\max(m_1, m_2, n)]/(nm_2)}$ established by [7]. In literature, [5] shows that there exists a local minimizer of the objective function whose estimation errors match ours. However, it is unknown if their estimator can achieve such convergence rate. On the other hand, our theorem confirms that our algorithm is able to find such estimator with optimal rate of convergence.

## 3.2 Estimation error in max norm and spectral norm

We next show the estimation error in max norm and spectral norm. Trivially, these estimation errors are bounded by that in Frobenius norm shown in Theorem 3.5. To develop improved rates of convergence in max and spectral norms, we need to impose stronger conditions on true parameters.

We first introduce some important notations. Denote $d_k$ as the maximum number of non-zeros in any row of the true precision matrices $\mathbf{\Omega}_k^*$, that is,

$$d_k := \max_{i \in \{1, \ldots, m_k\}} \left| \{ j \in \{1, \ldots, m_k\} : [\mathbf{\Omega}_k^*]_{i,j} \neq 0 \} \right|, \tag{3.7}$$

with $|\cdot|$ the cardinality of the inside set. For each covariance matrix $\mathbf{\Sigma}_k^*$, we define $\kappa_{\mathbf{\Sigma}_k^*} := \|\|\mathbf{\Sigma}_k^*\|\|_\infty$. Denote the Hessian matrix $\mathbf{\Gamma}_k^* := \mathbf{\Omega}_k^{*-1} \otimes \mathbf{\Omega}_k^{*-1} \in \mathbb{R}^{m_k^2 \times m_k^2}$, whose entry $[\mathbf{\Gamma}_k^*]_{(i,j),(s,t)}$ corresponds to the second order partial derivative of the objective function with respect to $[\mathbf{\Omega}_k]_{i,j}$ and $[\mathbf{\Omega}_k]_{s,t}$. We define its sub-matrix indexed by the index set $\mathbb{S}_k$ as $[\mathbf{\Gamma}_k^*]_{\mathbb{S}_k,\mathbb{S}_k} = [\mathbf{\Omega}_k^{*-1} \otimes \mathbf{\Omega}_k^{*-1}]_{\mathbb{S}_k,\mathbb{S}_k}$, which is the $|\mathbb{S}_k| \times |\mathbb{S}_k|$ matrix with rows and columns of $\mathbf{\Gamma}_k^*$ indexed by $\mathbb{S}_k$ and $\mathbb{S}_k$, respectively. Moreover, we define $\kappa_{\mathbf{\Gamma}_k^*} := \|\|([\mathbf{\Gamma}_k^*]_{\mathbb{S}_k,\mathbb{S}_k})^{-1}\|\|_\infty$. In order to establish the rate of convergence in max norm, we need to impose an irrepresentability condition on the Hessian matrix.

**Condition 3.7** (Irrepresentability). For each $k = 1, \ldots, K$, there exists some $\alpha_k \in (0, 1]$ such that

$$\max_{e \in \mathbb{S}_k^c} \left\| [\mathbf{\Gamma}_k^*]_{e,\mathbb{S}_k} \left( [\mathbf{\Gamma}_k^*]_{\mathbb{S}_k,\mathbb{S}_k} \right)^{-1} \right\|_1 \leq 1 - \alpha_k.$$

Condition 3.7 controls the influence of the non-connected terms in $\mathbb{S}_k^c$ on the connected edges in $\mathbb{S}_k$. This condition has been widely applied in lasso penalized models [26, 27].

**Condition 3.8** (Bounded Complexity). For each $k = 1, \ldots, K$, the parameters $\kappa_{\mathbf{\Sigma}_k^*}, \kappa_{\mathbf{\Gamma}_k^*}$ are bounded and the parameter $d_k$ in (3.7) satisfies $d_k = o\big(\sqrt{nm}/(m_k \log m_k)\big)$.

**Theorem 3.9.** Suppose Conditions 3.2, 3.3, 3.7 and 3.8 hold. Assume $s_k = O(m_k)$ for $k = 1, \ldots, K$ and assume $m_k's$ are in the same order, i.e., $m_1 \asymp m_2 \asymp \cdots \asymp m_K$. For each $k$, if the initialization satisfies $\mathbf{\Omega}_j^{(0)} \in \mathbb{B}(\mathbf{\Omega}_j^*)$ for any $j \neq k$, then the estimator $\widehat{\mathbf{\Omega}}_k$ from Algorithm 1 with $T = 2$ satisfies,

$$\left\| \widehat{\mathbf{\Omega}}_k - \mathbf{\Omega}_k^* \right\|_\infty = O_P \left( \sqrt{\frac{m_k \log m_k}{nm}} \right). \tag{3.8}$$

In addition, the edge set of $\widehat{\mathbf{\Omega}}_k$ is a subset of the true edge set of $\mathbf{\Omega}_k^*$, that is, $\text{supp}(\widehat{\mathbf{\Omega}}_k) \subseteq \text{supp}(\mathbf{\Omega}_k^*)$.

Theorem 3.9 shows that our Tlasso estimator achieves the optimal rate of convergence in max norm [4]. Here we consider the estimator obtained after two iterations since we require a new concentration inequality (Lemma B.3) for the sample covariance matrix, which is built upon the estimator in Theorem 3.5. A direct consequence from Theorem 3.9 is the estimation error in spectral norm.

**Corollary 3.10.** Suppose the conditions of Theorem 3.9 hold, for any $k = 1, \ldots, K$, we have

$$\left\| \widehat{\mathbf{\Omega}}_k - \mathbf{\Omega}_k^* \right\|_2 = O_P \left( d_k \sqrt{\frac{m_k \log m_k}{nm}} \right). \tag{3.9}$$

**Remark 3.11.** Now we compare our obtained rate of convergence in spectral norm for $K = 2$ with that established in the sparse matrix graphical model literature. In particular, [8] establishes the rate of $O_P\big(\sqrt{m_k(s_k \vee 1)\log(m_1 \vee m_2)/(nm_k)}\big)$ for $k = 1, 2$. Therefore, when $d_k^2 \leq (s_k \vee 1)$, which holds for example in the bounded degree graphs, our obtained rate is faster. However, our faster rate comes at the price of assuming the irrepresentability condition. Using recent advance in nonconvex regularization [28], we can eliminate the irrepresentability condition. We leave this to future work.

## 3.3 Model selection consistency

Theorem 3.9 ensures that the estimated precision matrix correctly excludes all non-informative edges and includes all the true edges $(i, j)$ with $|[\mathbf{\Omega}_k^*]_{i,j}| > C\sqrt{m_k \log m_k/(nm)}$ for some constant $C > 0$. Therefore, in order to achieve the model selection consistency, a sufficient condition is to assume that, for each $k = 1, \ldots, K$, the minimal signal $\theta_k := \min_{(i,j) \in \text{supp}(\mathbf{\Omega}_k^*)} [\mathbf{\Omega}_k^*]_{i,j}$ is not too small.

**Theorem 3.12.** Under the conditions of Theorem 3.9, if $\theta_k \geq C\sqrt{m_k \log m_k/(nm)}$ for some constant $C > 0$, then for any $k = 1, \ldots, K$, $\text{sign}(\widehat{\mathbf{\Omega}}_k) = \text{sign}(\mathbf{\Omega}_k^*)$, with high probability.

Theorem 3.12 indicates that our Tlasso estimator is able to correctly recover the graphical structure of each way of the high-dimensional tensor data. To the best of our knowledge, these is the first model selection consistency result in high dimensional tensor graphical model.

# 4 Simulations

We compare the proposed Tlasso estimator with two alternatives. The first one is the direct graphical lasso (Glasso) approach [21] which applies the glasso to the vectorized tensor data to estimate $\mathbf{\Omega}_1^* \otimes \cdots \otimes \mathbf{\Omega}_K^*$ directly. The second alternative method is the iterative penalized maximum likelihood method (P-MLE) proposed by [9], whose termination condition is set to be $\sum_{k=1}^{K} \big\| \widehat{\mathbf{\Omega}}_k^{(t)} - \widehat{\mathbf{\Omega}}_k^{(t-1)} \big\|_F \big/ K \leq 0.001$.

For simplicity, in our Tlasso algorithm we set the initialization of $k$-th precision matrix as $\mathbb{1}_{m_k}$ for each $k = 1, \ldots, K$ and the total iteration $T = 1$. The tuning parameter $\lambda_k$ is set as $20\sqrt{\log m_k/(nmm_k)}$. For a fair comparison, the same tuning parameter is applied in the P-MLE method. In the direct Glasso approach, its tuning parameter is chosen by cross-validation via *huge* package [29].

We consider two simulations with a third order tensor, i.e., $K = 3$. In Simulation 1, we construct a triangle graph, while in Simulation 2, we construct a four nearest neighbor graph for each precision matrix. An illustration of the generated graphs are shown in Figure 1. In each simulation, we consider three scenarios, i.e., s1: $n = 10$ and $(m_1, m_2, m_3) = (10, 10, 10)$; s2: $n = 50$ and $(m_1, m_2, m_3) = (10, 10, 10)$; s3: $n = 10$ and $(m_1, m_2, m_3) = (100, 5, 5)$. We repeat each example 100 times and compute the averaged computational time, the averaged estimation error of the Kronecker product of precision matrices $(m_1 m_2 m_3)^{-1} \big\| \widehat{\mathbf{\Omega}}_1 \otimes \cdots \otimes \widehat{\mathbf{\Omega}}_K - \mathbf{\Omega}_1^* \otimes \cdots \otimes \mathbf{\Omega}_K^* \big\|_F$, the true positive rate (TPR), and the true negative rate (TNR). More specifically, we denote $a_{i,j}^*$ be the $(i, j)$-th entry of $\mathbf{\Omega}_1^* \otimes \cdots \otimes \mathbf{\Omega}_K^*$, and define TPR $:= \sum_{i,j} \mathbb{1}(\widehat{a}_{i,j} \neq 0, a_{i,j}^* \neq 0) / \sum_{i,j} \mathbb{1}(a_{i,j}^* \neq 0)$ and TNR $:= \sum_{i,j} \mathbb{1}(\widehat{a}_{i,j} = 0, a_{i,j}^* = 0) / \sum_i \mathbb{1}(a_{i,j}^* = 0)$.

As shown in Figure 1, our Tlasso is dramatically faster than both alternative methods. In Scenario s3, Tlasso takes about five seconds for each replicate, the P-MLE takes about $500$ seconds while the direct Glasso method takes more than one hour and is omitted in the plot. Tlasso algorithm is not only computationally efficient but also enjoys superior estimation accuracy. In all examples, the direct Glasso method has significantly larger errors than Tlasso due to ignoring the tensor graphical structure. Tlasso outperforms P-MLE in Scenarios s1 and s2 and is comparable to it in Scenario s3.

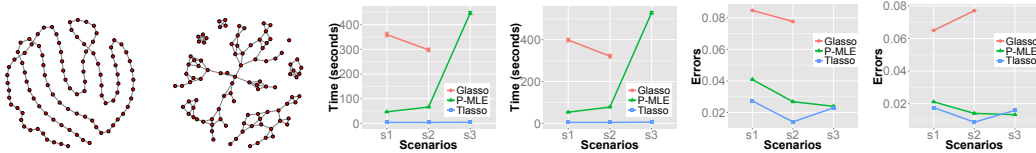

Figure 1: Left two plots: illustrations of the generated graphs; Middle two plots: computational time; Right two plots: estimation errors. In each group of two plots, the left (right) is for Simulation 1 (2).

Table 1 shows the variable selection performance. Our Tlasso identifies almost all edges in these six examples, while the Glasso and P-MLE method miss several true edges. On the other hand, Tlasso tends to include more non-connected edges than other methods.

Table 1: A comparison of variable selection performance. Here TPR and TNR denote the true positive rate and true negative rate.

| Scenarios | Glasso | | P-MLE | | Tlasso | |
|---|---|---|---|---|---|---|
| | TPR | TNR | TPR | TNR | TPR | TNR |
| s1 | 0.27 (0.002) | 0.96 (0.000) | 1 (0) | 0.89 (0.002) | 1(0) | 0.76 (0.004) |
| Sim 1 s2 | 0.34 (0.000) | 0.93 (0.000) | 1 (0) | 0.89 (0.002) | 1(0) | 0.76 (0.004) |
| s3 | / | / | 1 (0) | 0.93 (0.001) | 1(0) | 0.70 (0.004) |
| s1 | 0.08 (0.000) | 0.96 (0.000) | 0.93 (0.004) | 0.88 (0.002) | 1(0) | 0.65 (0.005) |
| Sim 2 s2 | 0.15 (0.000) | 0.92 (0.000) | 1 (0) | 0.85 (0.002) | 1(0) | 0.63 (0.005) |
| s3 | / | / | 0.82 (0.001) | 0.93 (0.001) | 0.99(0.001) | 0.38 (0.002) |

## Acknowledgement

We would like to thank the anonymous reviewers for their helpful comments. Han Liu is grateful for the support of NSF CAREER Award DMS1454377, NSF IIS1408910, NSF IIS1332109, NIH R01MH102339, NIH R01GM083084, and NIH R01HG06841. Guang Cheng's research is sponsored by NSF CAREER Award DMS1151692, NSF DMS1418042, Simons Fellowship in Mathematics, ONR N00014-15-1-2331 and a grant from Indiana Clinical and Translational Sciences Institute.

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
