[Supplementary Material]

# Non-convex Statistical Optimization for Sparse Tensor Graphical Model (Supplementary Material)

In this supplementary note, we provide the proofs of our main theorems in §A, prove the key lemmas in §B, list the auxiliary lemmas in §C, and illustrate additional simulation results in §D.

## A    Proof of main theorems

**Proof of Theorem 3.1:** To ease the presentation, we show that Theorem 3.1 holds when $K = 3$. The proof can be easily generalized to the case with $K > 3$.

We first simplify the population log-likelihood function. Note that when $\mathcal{T} \sim \text{TN}(\mathbf{0}; \mathbf{\Sigma}_1^*, \mathbf{\Sigma}_2^*, \mathbf{\Sigma}_3^*)$, Lemma 1 of [9] implies that $\text{vec}(\mathcal{T}) \sim \text{N}(\text{vec}(\mathbf{0}); \mathbf{\Sigma}_3^* \otimes \mathbf{\Sigma}_2^* \otimes \mathbf{\Sigma}_1^*)$. Therefore,

$$
\begin{aligned}
\mathbb{E}\big\{\text{tr}\big[\text{vec}(\mathcal{T})\text{vec}(\mathcal{T})^\top (\mathbf{\Omega}_3 \otimes \mathbf{\Omega}_2 \otimes \mathbf{\Omega}_1)\big]\big\} &= \text{tr}\big[(\mathbf{\Sigma}_3^* \otimes \mathbf{\Sigma}_2^* \otimes \mathbf{\Sigma}_1^*)(\mathbf{\Omega}_3 \otimes \mathbf{\Omega}_2 \otimes \mathbf{\Omega}_1)\big] \\
&= \text{tr}(\mathbf{\Sigma}_3^* \mathbf{\Omega}_3)\text{tr}(\mathbf{\Sigma}_2^* \mathbf{\Omega}_2)\text{tr}(\mathbf{\Sigma}_1^* \mathbf{\Omega}_1),
\end{aligned}
$$

where the second equality is due to the properties of kronecker product that $(\mathbf{A} \otimes \mathbf{B})(\mathbf{C} \otimes \mathbf{D}) = (\mathbf{AC}) \otimes (\mathbf{BD})$ and $\text{tr}(\mathbf{A} \otimes \mathbf{B}) = \text{tr}(\mathbf{A})\text{tr}(\mathbf{B})$. Therefore, the population log-likelihood function can be rewritten as

$$
q(\mathbf{\Omega}_1, \mathbf{\Omega}_2, \mathbf{\Omega}_3) = \frac{\text{tr}(\mathbf{\Sigma}_3^* \mathbf{\Omega}_3)\text{tr}(\mathbf{\Sigma}_2^* \mathbf{\Omega}_2)\text{tr}(\mathbf{\Sigma}_1^* \mathbf{\Omega}_1)}{m_1 m_2 m_3} - \frac{1}{m_1}\log|\mathbf{\Omega}_1| - \frac{1}{m_2}\log|\mathbf{\Omega}_2| - \frac{1}{m_3}\log|\mathbf{\Omega}_3|.
$$

Taking derivative of $q(\mathbf{\Omega}_1, \mathbf{\Omega}_2, \mathbf{\Omega}_3)$ with respect to $\mathbf{\Omega}_1$ while fixing $\mathbf{\Omega}_2$ and $\mathbf{\Omega}_3$, we have

$$
\nabla_1 q(\mathbf{\Omega}_1, \mathbf{\Omega}_2, \mathbf{\Omega}_3) = \frac{\text{tr}(\mathbf{\Sigma}_3^* \mathbf{\Omega}_3)\text{tr}(\mathbf{\Sigma}_2^* \mathbf{\Omega}_2)}{m_1 m_2 m_3} \mathbf{\Sigma}_1^* - \frac{1}{m_1}\mathbf{\Omega}_1^{-1}.
$$

Setting it as zero leads to $\mathbf{\Omega}_1 = m_2 m_3 [\text{tr}(\mathbf{\Sigma}_3^* \mathbf{\Omega}_3)\text{tr}(\mathbf{\Sigma}_2^* \mathbf{\Omega}_2)]^{-1}\mathbf{\Omega}_1^*$. This is indeed a minimizer of $q(\mathbf{\Omega}_1, \mathbf{\Omega}_2, \mathbf{\Omega}_3)$ when fixing $\mathbf{\Omega}_2$ and $\mathbf{\Omega}_3$, since the second derivative $\nabla_1^2 q(\mathbf{\Omega}_1, \mathbf{\Omega}_2, \mathbf{\Omega}_3) = m_1^{-1}\mathbf{\Omega}_1^{-1} \otimes \mathbf{\Omega}_1^{-1}$ is positive definite. Therefore, we have

$$
M_1(\mathbf{\Omega}_2, \mathbf{\Omega}_3) = \frac{m_2 m_3}{\text{tr}(\mathbf{\Sigma}_3^* \mathbf{\Omega}_3)\text{tr}(\mathbf{\Sigma}_2^* \mathbf{\Omega}_2)}\mathbf{\Omega}_1^*. \tag{A.1}
$$

Therefore, $M_1(\mathbf{\Omega}_2, \mathbf{\Omega}_3)$ equals to the true parameter $\mathbf{\Omega}_1^*$ up to a constant. The computations of $M_2(\mathbf{\Omega}_1, \mathbf{\Omega}_3)$ and $M_3(\mathbf{\Omega}_1, \mathbf{\Omega}_2)$ follow from the same argument. This ends the proof of Theorem 3.1. ∎

**Proof of Theorem 3.4:** To ease the presentation, we show that (3.5) holds when $K = 3$. The proof of the case when $K > 3$ is similar. We focus on the proof of the statistical error for the sample minimization function $\widehat{M}_1(\mathbf{\Omega}_2, \mathbf{\Omega}_3)$.

By definition, $\widehat{M}_1(\mathbf{\Omega}_2, \mathbf{\Omega}_3) = \text{argmin}_{\mathbf{\Omega}_1} q_n(\mathbf{\Omega}_1, \mathbf{\Omega}_2, \mathbf{\Omega}_3) = \text{argmin}_{\mathbf{\Omega}_1} L(\mathbf{\Omega}_1)$, where

$$
L(\mathbf{\Omega}_1) = \frac{1}{m_1}\text{tr}(\mathbf{S}_1 \mathbf{\Omega}_1) - \frac{1}{m_1}\log|\mathbf{\Omega}_1| + \lambda_1 \|\mathbf{\Omega}_1\|_{1,\text{off}},
$$

with the sample covariance matrix

$$
\mathbf{S}_1 = \frac{1}{m_2 m_3 n}\sum_{i=1}^n \mathbf{V}_i \mathbf{V}_i^\top \text{ with } \mathbf{V}_i = \big[\mathcal{T}_i \times \{\mathbb{1}_{m_1}, \mathbf{\Omega}_2^{1/2}, \mathbf{\Omega}_3^{1/2}\}\big]_{(1)}.
$$

For some constant $H > 0$, we define the set of convergence

$$
\mathbb{A} := \left\{\mathbf{\Delta} \in \mathbb{R}^{m_1 \times m_1} : \mathbf{\Delta} = \mathbf{\Delta}^\top, \|\mathbf{\Delta}\|_F = H\sqrt{\frac{(m_1 + s_1)\log m_1}{n m_2 m_3}}\right\}.
$$

The key idea is to show that

$$
\inf_{\mathbf{\Delta} \in \mathbb{A}} \big\{L\big(M_1(\mathbf{\Omega}_2, \mathbf{\Omega}_3) + \mathbf{\Delta}\big) - L\big(M_1(\mathbf{\Omega}_2, \mathbf{\Omega}_3)\big)\big\} > 0, \tag{A.2}
$$

with high probability. To understand it, note that the function $L\big(M_1(\boldsymbol{\Omega}_2,\boldsymbol{\Omega}_3)+\boldsymbol{\Delta}\big)-L\big(M_1(\boldsymbol{\Omega}_2,\boldsymbol{\Omega}_3)\big)$ is convex in $\boldsymbol{\Delta}$. In addition, since $\widehat{M}_1(\boldsymbol{\Omega}_2,\boldsymbol{\Omega}_3)$ minimizes $L(\boldsymbol{\Omega}_1)$, we have

$$L\big(\widehat{M}_1(\boldsymbol{\Omega}_2,\boldsymbol{\Omega}_3)\big)-L\big(M_1(\boldsymbol{\Omega}_2,\boldsymbol{\Omega}_3)\big)\le L\big(M_1(\boldsymbol{\Omega}_2,\boldsymbol{\Omega}_3)\big)-L\big(M_1(\boldsymbol{\Omega}_2,\boldsymbol{\Omega}_3)\big)=0.$$

If we can show (A.2), then the minimizer $\widehat{\boldsymbol{\Delta}}=\widehat{M}_1(\boldsymbol{\Omega}_2,\boldsymbol{\Omega}_3)-M_1(\boldsymbol{\Omega}_2,\boldsymbol{\Omega}_3)$ must be within the interior of the ball defined by $\mathbb{A}$, and hence $\|\widehat{\boldsymbol{\Delta}}\|_F\le H\sqrt{(m_1+s_1)\log m_1/(nm_2m_3)}$. Similar technique is applied in vector-valued graphical model literature [25].

To show (A.2), we first decompose $L\big(M_1(\boldsymbol{\Omega}_2,\boldsymbol{\Omega}_3)+\boldsymbol{\Delta}\big)-L\big(M_1(\boldsymbol{\Omega}_2,\boldsymbol{\Omega}_3)\big)=I_1+I_2+I_3$, where

$$
\begin{aligned}
I_1 &:= \frac{1}{m_1}\operatorname{tr}(\boldsymbol{\Delta}\mathbf{S}_1)-\frac{1}{m_1}\big\{\log|M_1(\boldsymbol{\Omega}_2,\boldsymbol{\Omega}_3)+\boldsymbol{\Delta}|-\log|M_1(\boldsymbol{\Omega}_2,\boldsymbol{\Omega}_3)|\big\},\\
I_2 &:= \lambda_1\big\{\|[M_1(\boldsymbol{\Omega}_2,\boldsymbol{\Omega}_3)+\boldsymbol{\Delta}]_{\mathbb{S}_1}\|_1-\|[M_1(\boldsymbol{\Omega}_2,\boldsymbol{\Omega}_3)]_{\mathbb{S}_1}\|_1\big\},\\
I_3 &:= \lambda_1\big\{\|[M_1(\boldsymbol{\Omega}_2,\boldsymbol{\Omega}_3)+\boldsymbol{\Delta}]_{\mathbb{S}_1^c}\|_1-\|[M_1(\boldsymbol{\Omega}_2,\boldsymbol{\Omega}_3)]_{\mathbb{S}_1^c}\|_1\big\}.
\end{aligned}
$$

It is sufficient to show $I_1+I_2+I_3>0$ with high probability. To simplify the term $I_1$, we employ the Taylor expansion of $f(t)=\log|M_1(\boldsymbol{\Omega}_2,\boldsymbol{\Omega}_3)+t\boldsymbol{\Delta}|$ at $t=0$ to obtain

$$\log|M_1(\boldsymbol{\Omega}_2,\boldsymbol{\Omega}_3)+\boldsymbol{\Delta}|-\log|M_1(\boldsymbol{\Omega}_2,\boldsymbol{\Omega}_3)|$$

$$=\operatorname{tr}\big\{[M_1(\boldsymbol{\Omega}_2,\boldsymbol{\Omega}_3)]^{-1}\boldsymbol{\Delta}\big\}-[\operatorname{vec}(\boldsymbol{\Delta})]^\top\left[\int_0^1(1-\nu)\mathbf{M}_\nu^{-1}\otimes\mathbf{M}_\nu^{-1}\mathrm{d}\nu\right]\operatorname{vec}(\boldsymbol{\Delta}),$$

where $\mathbf{M}_\nu:=M_1(\boldsymbol{\Omega}_2,\boldsymbol{\Omega}_3)+\nu\boldsymbol{\Delta}\in\mathbb{R}^{m_1\times m_1}$. This leads to

$$I_1=\underbrace{\frac{1}{m_1}\operatorname{tr}\big(\{\mathbf{S}_1-[M_1(\boldsymbol{\Omega}_2,\boldsymbol{\Omega}_3)]^{-1}\}\boldsymbol{\Delta}\big)}_{I_{11}}+\underbrace{\frac{1}{m_1}[\operatorname{vec}(\boldsymbol{\Delta})]^\top\left[\int_0^1(1-\nu)\mathbf{M}_\nu^{-1}\otimes\mathbf{M}_\nu^{-1}\mathrm{d}\nu\right]\operatorname{vec}(\boldsymbol{\Delta})}_{I_{12}}.$$

For two symmetric matrices $\mathbf{A},\mathbf{B}$, it is easy to see that $|\operatorname{tr}(\mathbf{A}\mathbf{B})|=|\sum_{i,j}\mathbf{A}_{i,j}\mathbf{B}_{i,j}|$. Based on this observation, we decompose $I_{11}$ into two parts: those in the set $\mathbb{S}_1=\{(i,j):[\boldsymbol{\Omega}_1^*]_{i,j}\ne0\}$ and those not in $\mathbb{S}_1$. That is, $|I_{11}|\le I_{111}+I_{112}$, where

$$
\begin{aligned}
I_{111} &:= \frac{1}{m_1}\Big|\sum_{(i,j)\in\mathbb{S}_1}\big\{\mathbf{S}_1-[M_1(\boldsymbol{\Omega}_2,\boldsymbol{\Omega}_3)]^{-1}\big\}_{i,j}\boldsymbol{\Delta}_{i,j}\Big|,\\
I_{112} &:= \frac{1}{m_1}\Big|\sum_{(i,j)\notin\mathbb{S}_1}\big\{\mathbf{S}_1-[M_1(\boldsymbol{\Omega}_2,\boldsymbol{\Omega}_3)]^{-1}\big\}_{i,j}\boldsymbol{\Delta}_{i,j}\Big|.
\end{aligned}
$$

**Bound $I_{111}$:** For two matrices $\mathbf{A},\mathbf{B}$ and a set $\mathbb{S}$, we have

$$\Big|\sum_{(i,j)\in\mathbb{S}}\mathbf{A}_{i,j}\mathbf{B}_{i,j}\Big|\le\max_{i,j}|\mathbf{A}_{i,j}|\Big|\sum_{(i,j)\in\mathbb{S}}\mathbf{B}_{i,j}\Big|\le\sqrt{|\mathbb{S}|}\max_{i,j}|\mathbf{A}_{i,j}|\|\mathbf{B}\|_F,$$

where the second inequality is due to the Cauchy-Schwarz inequality and the fact that $\sum_{(i,j)\in\mathbb{S}}\mathbf{B}_{i,j}^2\le\|\mathbf{B}\|_F^2$. Therefore, we have

$$
\begin{aligned}
I_{111} &\le \frac{\sqrt{s_1+m_1}}{m_1}\cdot\max_{i,j}\Big|\big\{\mathbf{S}_1-[M_1(\boldsymbol{\Omega}_2,\boldsymbol{\Omega}_3)]^{-1}\big\}_{ij}\Big|\|\boldsymbol{\Delta}\|_F\\
&\le C\sqrt{\frac{(m_1+s_1)\log m_1}{nm_1^2m_2m_3}}\|\boldsymbol{\Delta}\|_F=\frac{CH\cdot(m_1+s_1)\log m_1}{nm_1m_2m_3},
\end{aligned}
\tag{A.3}
$$

where (A.3) is from Lemma B.2, the definition of $M_1(\boldsymbol{\Omega}_2,\boldsymbol{\Omega}_3)$ in (A.1), and the fact that $\boldsymbol{\Delta}\in\mathbb{A}$.

**Bound $I_{12}$:** For any vector $\mathbf{v}\in\mathbb{R}^p$ and any matrix $\mathbf{A}\in\mathbb{R}^{p\times p}$, the variational form of Rayleigh quotients implies $\lambda_{\min}(\mathbf{A})=\min_{\|\mathbf{x}\|=1}\mathbf{x}^\top\mathbf{A}\mathbf{x}$ and hence $\lambda_{\min}(\mathbf{A})\|\mathbf{v}\|^2\le\mathbf{v}^\top\mathbf{A}\mathbf{v}$. Setting $\mathbf{v}=\operatorname{vec}(\boldsymbol{\Delta})$ and $\mathbf{A}=\int_0^1(1-\nu)\mathbf{M}_\nu^{-1}\otimes\mathbf{M}_\nu^{-1}\mathrm{d}\nu$ leads to

$$I_{12}\ge\frac{1}{m_1}\|\operatorname{vec}(\boldsymbol{\Delta})\|_2^2\int_0^1(1-\nu)\lambda_{\min}\big(\mathbf{M}_\nu^{-1}\otimes\mathbf{M}_\nu^{-1}\big)\,\mathrm{d}\nu.$$

Moreover, by the property of kronecker product, we have

$$\lambda_{\min}\left(\mathbf{M}_\nu^{-1} \otimes \mathbf{M}_\nu^{-1}\right) = [\lambda_{\min}(\mathbf{M}_\nu^{-1})]^2 = [\lambda_{\max}(\mathbf{M}_\nu)]^{-2}.$$

In addition, by definition, $\mathbf{M}_\nu = M_1(\mathbf{\Omega}_2, \mathbf{\Omega}_3) + \nu\mathbf{\Delta}$, and hence we have

$$\lambda_{\max}[M_1(\mathbf{\Omega}_2, \mathbf{\Omega}_3) + \nu\mathbf{\Delta}] \le \lambda_{\max}[M_1(\mathbf{\Omega}_2, \mathbf{\Omega}_3)] + \lambda_{\max}(\nu\mathbf{\Delta}).$$

Therefore, we can bound $I_{12}$ from below, that is,

$$
\begin{aligned}
I_{12} &\ge \frac{\|\mathrm{vec}(\mathbf{\Delta})\|_2^2}{2m_1} \min_{0 \le \nu \le 1} \left[\lambda_{\max}[M_1(\mathbf{\Omega}_2, \mathbf{\Omega}_3)] + \lambda_{\max}(\nu\mathbf{\Delta})\right]^{-2} \\
&\ge \frac{\|\mathrm{vec}(\mathbf{\Delta})\|_2^2}{2m_1} \left[\|M_1(\mathbf{\Omega}_2, \mathbf{\Omega}_3)\|_2 + \|\mathbf{\Delta}\|_2\right]^{-2}.
\end{aligned}
$$

On the boundary of $\mathbb{A}$, it holds that $\|\mathbf{\Delta}\|_2 \le \|\mathbf{\Delta}\|_F = o(1)$. Moreover, according to (A.1), we have

$$\|M_1(\mathbf{\Omega}_2, \mathbf{\Omega}_3)\|_2 = \left|\frac{m_2 m_3}{\mathrm{tr}(\mathbf{\Sigma}_3^* \mathbf{\Omega}_3)\mathrm{tr}(\mathbf{\Sigma}_2^* \mathbf{\Omega}_2)}\right| \|\mathbf{\Omega}_1^*\|_2 \le \frac{100}{81}\|\mathbf{\Sigma}_1^*\|_2 \le \frac{1.5}{C_1}, \qquad (A.4)$$

where the first inequality is due to

$$
\begin{aligned}
\mathrm{tr}(\mathbf{\Sigma}_3^* \mathbf{\Omega}_3) &= \mathrm{tr}[\mathbf{\Sigma}_3^*(\mathbf{\Omega}_3 - \mathbf{\Omega}_3^*) + \mathbb{1}_{m_3}] \ge m_3 - |\mathrm{tr}[\mathbf{\Sigma}_3^*(\mathbf{\Omega}_3 - \mathbf{\Omega}_3^*)]| \\
&\ge m_3 - \|\mathbf{\Sigma}_3^*\|_F \|\mathbf{\Omega}_3 - \mathbf{\Omega}_3^*\|_F \ge m_3(1 - \alpha\|\mathbf{\Sigma}_3^*\|_2/\sqrt{m_3}) \ge 0.9 m_3,
\end{aligned}
$$

for sufficiently large $m_3$. Similarly, it holds that $\mathrm{tr}(\mathbf{\Sigma}_2^* \mathbf{\Omega}_2) \ge 0.9 m_2$. The second inequality in (A.4) is due to Condition 3.2. This together with the fact that $\|\mathrm{vec}(\mathbf{\Delta})\|_2 = \|\mathbf{\Delta}\|_F = o(1) \le 0.5/C_1$ for sufficiently large $n$ imply that

$$I_{12} \ge \frac{\|\mathrm{vec}(\mathbf{\Delta})\|_2^2}{2m_1}\left(\frac{C_1}{2}\right)^2 = \frac{C_1^2 H^2}{8} \cdot \frac{(m_1 + s_1)\log m_1}{n m_1 m_2 m_3}, \qquad (A.5)$$

which dominates the term $I_{111}$ for sufficiently large $H$.

**Bound $I_2$:** To bound $I_2$, we apply the triangle inequality and then connect the $\ell_1$ matrix norm with its Frobenius norm to obtain the final bound. Specifically, we have

$$|I_2| \le \lambda_1 \|[\mathbf{\Delta}]_{\mathbb{S}_1}\|_1 = \lambda_1 \sum_{(i,j)\in\mathbb{S}_1} |\mathbf{\Delta}_{i,j}| \le \lambda_1 \sqrt{(s_1 + m_1)\sum_{(i,j)\in\mathbb{S}_1} \mathbf{\Delta}_{i,j}^2} \le \lambda_1 \sqrt{s_1 + m_1}\|\mathbf{\Delta}\|_F,$$

where the first inequality is from triangle inequality, the second inequality is due to the Cauchy-Schwarz inequality by noting that $s_1 = |\mathbb{S}_1| - m_1$, and the last inequality is due to the definition of Frobenius norm. By Condition 3.3, $\lambda_1 \le C_2\sqrt{\log m_1/(n m_1^2 m_2 m_3)}$. Therefore,

$$|I_2| \le C_2 H \cdot \frac{(m_1 + s_1)\log m_1}{n m_1 m_2 m_3},$$

which is dominated by $I_{12}$ for sufficiently large $H$ according to (A.5).

**Bound $I_3 - |I_{112}|$:** We show $I_3 - |I_{112}| > 0$. According to (A.1), we have that $M_1(\mathbf{\Omega}_2, \mathbf{\Omega}_3)$ equals $\mathbf{\Omega}_1^*$ up to a non-zero coefficient. Therefore, for any entry $(i,j) \in \mathbb{S}_1^c$, we have $[M_1(\mathbf{\Omega}_2, \mathbf{\Omega}_3)]_{i,j} = 0$. This implies that

$$I_3 = \lambda_1 \sum_{(i,j)\in\mathbb{S}_1^c} \left\{|[M_1(\mathbf{\Omega}_2, \mathbf{\Omega}_3)]_{i,j} + \mathbf{\Delta}_{i,j}| - |[M_1(\mathbf{\Omega}_2, \mathbf{\Omega}_3)]_{i,j}|\right\} = \lambda_1 \sum_{(i,j)\in\mathbb{S}_1^c} |\mathbf{\Delta}_{i,j}|.$$

This together with the expression of $I_{112}$ and the bound in Lemma B.2 leads to

$$
\begin{aligned}
I_3 - I_{112} &= \sum_{(i,j)\in\mathbb{S}_1^c} \left\{\lambda_1 - m_1^{-1}\left\{\mathbf{S}_1 - [M_1(\mathbf{\Omega}_2, \mathbf{\Omega}_3)]^{-1}\right\}_{i,j}\right\} |\mathbf{\Delta}_{i,j}| \\
&\ge \left(\lambda_1 - C\sqrt{\frac{\log m_1}{n m_1^2 m_2 m_3}}\right) \sum_{(i,j)\in\mathbb{S}_1^c} |\mathbf{\Delta}_{i,j}| > 0,
\end{aligned}
$$

as long as $1/C_2 > C$ for some constant $C$, which is valid for sufficient small $C_2$ in Condition 3.3.

Combining all these bounds together, we have, for any $\mathbf{\Delta} \in \mathbb{A}$, with high probability,

$$L\big(M_1(\mathbf{\Omega}_2, \mathbf{\Omega}_3) + \mathbf{\Delta}\big) - L\big(M_1(\mathbf{\Omega}_2, \mathbf{\Omega}_3)\big) \geq I_{12} - I_{111} - |I_2| + I_3 - I_{112} > 0,$$

which ends the proof Theorem 3.4. $\blacksquare$

**Proof of Theorem 3.5:** We show it by connecting the one-step convergence result in Theorem 3.1 and the statistical error result in Theorem 3.4. We show the case when $K = 3$. The proof of the $K > 3$ case is similar. We focus on the proof of the estimation error $\big\|\widehat{\mathbf{\Omega}}_1 - \mathbf{\Omega}_1^*\big\|_F$.

To ease the presentation, in the following derivation we remove the superscript in the initializations $\mathbf{\Omega}_2^{(0)}$ and $\mathbf{\Omega}_3^{(0)}$ and use $\mathbf{\Omega}_2$ and $\mathbf{\Omega}_3$ instead. According to the procedure in Algorithm 1, we have

$$
\begin{aligned}
\big\|\widehat{\mathbf{\Omega}}_1 - \mathbf{\Omega}_1^*\big\|_F &= \left\| \frac{\widehat{M}_1(\mathbf{\Omega}_2, \mathbf{\Omega}_3)}{\big\|\widehat{M}_1(\mathbf{\Omega}_2, \mathbf{\Omega}_3)\big\|_F} - \frac{\widehat{M}_1(\mathbf{\Omega}_2, \mathbf{\Omega}_3)}{\big\|\widehat{M}_1(\mathbf{\Omega}_2, \mathbf{\Omega}_3)\big\|_F} \right\|_F \\
&\leq \left\| \frac{\widehat{M}_1(\mathbf{\Omega}_2, \mathbf{\Omega}_3)}{\big\|\widehat{M}_1(\mathbf{\Omega}_2, \mathbf{\Omega}_3)\big\|_F} - \frac{M_1(\mathbf{\Omega}_2, \mathbf{\Omega}_3)}{\big\|\widehat{M}_1(\mathbf{\Omega}_2, \mathbf{\Omega}_3)\big\|_F} \right\|_F + \left\| \frac{M_1(\mathbf{\Omega}_2, \mathbf{\Omega}_3)}{\big\|\widehat{M}_1(\mathbf{\Omega}_2, \mathbf{\Omega}_3)\big\|_F} - \frac{M_1(\mathbf{\Omega}_2, \mathbf{\Omega}_3)}{\big\|M_1(\mathbf{\Omega}_2, \mathbf{\Omega}_3)\big\|_F} \right\|_F \\
&\leq \frac{2}{\big\|\widehat{M}_1(\mathbf{\Omega}_2, \mathbf{\Omega}_3)\big\|_F} \big\|\widehat{M}_1(\mathbf{\Omega}_2, \mathbf{\Omega}_3) - M_1(\mathbf{\Omega}_2, \mathbf{\Omega}_3)\big\|_F,
\end{aligned}
$$

where the last inequality is due to the triangle inequality $||a| - |b|| \leq |a - b|$ and the summation of two parts. We next bound $\big\|\widehat{M}_1(\mathbf{\Omega}_2, \mathbf{\Omega}_3)\big\|_F$. By triangle inequality,

$$\big\|\widehat{M}_1(\mathbf{\Omega}_2, \mathbf{\Omega}_3)\big\|_F \geq \|M_1(\mathbf{\Omega}_2, \mathbf{\Omega}_3)\|_F - \big\|M_1(\mathbf{\Omega}_2, \mathbf{\Omega}_3) - \widehat{M}_1(\mathbf{\Omega}_2, \mathbf{\Omega}_3)\big\|_F \geq 2^{-1}\|M_1(\mathbf{\Omega}_2, \mathbf{\Omega}_3)\|_F,$$

since $\big\|M_1(\mathbf{\Omega}_2, \mathbf{\Omega}_3) - \widehat{M}_1(\mathbf{\Omega}_2, \mathbf{\Omega}_3)\big\|_F = o_P(1)$ as shown in Theorem 3.4. Moreover, by the Cauchy-Schwarz inequality, we have

$$\mathrm{tr}(\mathbf{\Sigma}_2^* \mathbf{\Omega}_2) \leq \|\mathbf{\Sigma}_2^*\|_F \|\mathbf{\Omega}_2\|_F \leq m_2 \|\mathbf{\Sigma}_2^*\|_2 \|\mathbf{\Omega}_2\|_2 \leq 2m_2/C_1,$$

due to Condition 3.2 and the fact that $\mathbf{\Omega}_2 \in \mathbb{B}(\mathbf{\Omega}_2^*)$. Similarly, we have $\mathrm{tr}(\mathbf{\Sigma}_3^* \mathbf{\Omega}_3) \leq 2m_3/C_1$. This together with the expression of $M_1(\mathbf{\Omega}_2, \mathbf{\Omega}_3)$ in (A.1) imply that $\big\|\widehat{M}_1(\mathbf{\Omega}_2, \mathbf{\Omega}_3)\big\|_F \geq C_1^2/4$ and hence

$$\big\|\widehat{\mathbf{\Omega}}_1 - \mathbf{\Omega}_1^*\big\|_F \leq \frac{8}{C_1^2} \big\|\widehat{M}_1(\mathbf{\Omega}_2, \mathbf{\Omega}_3) - M_1(\mathbf{\Omega}_2, \mathbf{\Omega}_3)\big\|_F = O_P\left(\sqrt{\frac{m_1(m_1 + s_1)\log m_1}{nm_1 m_2 m_3}}\right),$$

according to Theorem 3.4. This ends the proof Theorem 3.5. $\blacksquare$

**Proof of Theorem 3.9:** We prove it by transferring the optimization problem to an equivalent primal-dual problem and then applying the convergence results of [27] to obtain the desirable rate of convergence.

Given the sample covariance matrix $\widehat{\mathbf{S}}_k$ defined in Lemma B.3, according to (2.3), for each $k = 1, \ldots, K$, the optimization problem has a unique solution $\widehat{\mathbf{\Omega}}_k$ which satisfies the following Karush-Kuhn-Tucker (KKT) conditions

$$\widehat{\mathbf{S}}_k - \widehat{\mathbf{\Omega}}_k + m_k \lambda_k \widehat{\mathbf{Z}}_k = 0, \tag{A.6}$$

where $\widehat{\mathbf{Z}}_k \in \mathbb{R}^{m_k \times m_k}$ belongs to the sub-differential of $\|\mathbf{\Omega}_k\|_{1,\mathrm{off}}$ evaluated at $\widehat{\mathbf{\Omega}}_k$, that is,

$$[\widehat{\mathbf{Z}}_k]_{i,j} := \begin{cases} 0, & \text{if } i = j \\ \mathrm{sign}([\widehat{\mathbf{\Omega}}_k]_{i,j}) & \text{if } i \neq j \text{ and } [\widehat{\mathbf{\Omega}}_k]_{i,j} \neq 0 \\ \in [-1, +1] & \text{if } i \neq j \text{ and } [\widehat{\mathbf{\Omega}}_k]_{i,j} = 0. \end{cases}$$

Following [27], we construct the primary-dual witness solution $(\widetilde{\mathbf{\Omega}}_k, \widetilde{\mathbf{Z}}_k)$ such that

$$\widetilde{\mathbf{\Omega}}_k := \operatorname*{argmin}_{\mathbf{\Omega}_k \succ 0, \mathbf{\Omega}_k = \mathbf{\Omega}_k^\top, [\mathbf{\Omega}_k]_{\mathbb{S}_k^c} = 0} \left\{ \mathrm{tr}(\widehat{\mathbf{S}}_k \mathbf{\Omega}_k) - \log|\mathbf{\Omega}_k| + m_k \lambda_k \|\mathbf{\Omega}_k\|_{1,\mathrm{off}} \right\},$$

where the set $\mathbb{S}_k$ refers to the set of true non-zero edges of $\boldsymbol{\Omega}_k^*$. Therefore, by construction, the support of the dual estimator $\widetilde{\boldsymbol{\Omega}}_k$ is a subset of the true support, i.e., $\mathrm{supp}(\widetilde{\boldsymbol{\Omega}}_k) \subseteq \mathrm{supp}(\boldsymbol{\Omega}_k^*)$. We then construct $\widetilde{\mathbf{Z}}_k$ as the sub-differential $\widehat{\mathbf{Z}}_k$ and then for each $(i,j) \in \mathbb{S}_k^c$, we replace $[\widetilde{\mathbf{Z}}_k]_{i,j}$ with $([\widetilde{\boldsymbol{\Omega}}_k^{-1}]_{i,j} - [\widehat{\mathbf{S}}_k]_{i,j})/(m_k\lambda_k)$ to ensure that $(\widetilde{\boldsymbol{\Omega}}_k, \widetilde{\mathbf{Z}}_k)$ satisfies the optimality condition (A.6).

Denote $\boldsymbol{\Delta} := \widetilde{\boldsymbol{\Omega}}_k - \boldsymbol{\Omega}_k^*$ and $R(\boldsymbol{\Delta}) := \widetilde{\boldsymbol{\Omega}}_k^{-1} - \boldsymbol{\Omega}_k^{*-1} + \boldsymbol{\Omega}_k^{*-1}\boldsymbol{\Delta}\widetilde{\boldsymbol{\Omega}}_k^{-1}$. According to Lemma 4 of [27], in order to show the strict dual feasibility $\widetilde{\boldsymbol{\Omega}}_k = \widehat{\boldsymbol{\Omega}}_k$, it is sufficient to prove

$$\max\{\|\widehat{\mathbf{S}}_k - \boldsymbol{\Sigma}_k^*\|_\infty, \|R(\boldsymbol{\Delta})\|_\infty\} \le \frac{\alpha_k m_k \lambda_k}{8},$$

with $\alpha_k$ defined in Condition 3.7. As assumed in Condition 3.3, the tuning parameter satisfies $1/C_2\sqrt{\log m_k/(nmm_k)} \le \lambda_k \le C_2\sqrt{\log m_k/(nmm_k)}$ for some constant $C_2 > 0$ and hence $\alpha_k m_k \lambda_k/8 \ge C_3\sqrt{m_k \log m_k/(nm)}$ for some constant $C_3 > 0$.

In addition, according to Lemma B.3, we have

$$\|\widehat{\mathbf{S}}_k - \boldsymbol{\Sigma}_k^*\|_\infty = O_P\left(\max_{j=1,\dots,K}\sqrt{\frac{(m_j + s_j)\log m_j}{nm}}\right).$$

Under the assumption that $s_j = O(m_j)$ for $j = 1, \dots, K$ and $m_1 \asymp m_2 \asymp \cdots \asymp m_K$, we have

$$\|\widehat{\mathbf{S}}_k - \boldsymbol{\Sigma}_k^*\|_\infty = O_P\left(\sqrt{\frac{m_k \log m_k}{nm}}\right).$$

Therefore, there exists a sufficiently small constant $C_2$ such that $\|\widehat{\mathbf{S}}_k - \boldsymbol{\Sigma}_k^*\|_\infty \le \alpha_k m_k \lambda_k/8$.

Moreover, according to Lemma 5 of [27], $\|R(\boldsymbol{\Delta})\|_\infty \le 1.5 d_k \|\boldsymbol{\Delta}\|_\infty^2 \kappa_{\boldsymbol{\Sigma}_k^*}^3$ as long as $\|\boldsymbol{\Delta}\|_\infty \le (3\kappa_{\boldsymbol{\Sigma}_k^*} d_k)^{-1}$. According to Lemma 6 of [27], if we can show

$$r := 2\kappa_{\boldsymbol{\Gamma}_k^*}\left(\|\widehat{\mathbf{S}}_k - \boldsymbol{\Sigma}_k^*\|_\infty + m_k\lambda_k\right) \le \min\left\{\frac{1}{3\kappa_{\boldsymbol{\Sigma}_k^*} d_k}, \frac{1}{\kappa_{\boldsymbol{\Sigma}_k^*}^3 \kappa_{\boldsymbol{\Gamma}_k^*} d_k}\right\},$$

then we have $\|\boldsymbol{\Delta}\|_\infty \le r$. By Condition 3.8, $\kappa_{\boldsymbol{\Gamma}_k^*}$ and $\kappa_{\boldsymbol{\Sigma}_k^*}$ are bounded. Therefore, $\|\widehat{\mathbf{S}}_k - \boldsymbol{\Sigma}_k^*\|_\infty + m_k\lambda_k$ is in the same order of $\sqrt{m_k \log m_k/(nm)}$, which is in a smaller order of $d_k^{-1}$ by the assumption of $d_k$ in Condition 3.8. Therefore, we have shown that $\|R(\boldsymbol{\Delta})\|_\infty \le m_k\lambda_k$ for a sufficiently small constant $C_2$.

Combining above two bounds, we achieve the strict dual feasibility $\widetilde{\boldsymbol{\Omega}}_k = \widehat{\boldsymbol{\Omega}}_k$. Therefore, we have $\mathrm{supp}(\widehat{\boldsymbol{\Omega}}_k) \subseteq \mathrm{supp}(\boldsymbol{\Omega}_k^*)$ and moreover,

$$\|\widehat{\boldsymbol{\Omega}}_k - \boldsymbol{\Omega}_k^*\|_\infty = \|\boldsymbol{\Delta}\|_\infty = O_P\left(\sqrt{\frac{m_k \log m_k}{nm}}\right).$$

This ends the proof of Theorem 3.9. ∎

# B  Proof of key lemmas

The first key lemma establishes the rate of convergence of the difference between a sample-based quadratic form and its expectation. This new concentration result is also of independent interest.

**Lemma B.1.** Assume i.i.d. data $\mathbf{X}, \mathbf{X}_1, \dots, \mathbf{X}_n \in \mathbb{R}^{p \times q}$ follows the matrix-variate normal distribution such that $\mathrm{vec}(\mathbf{X}_i) \sim \mathrm{N}(\mathbf{0}; \boldsymbol{\Psi}^* \otimes \boldsymbol{\Sigma}^*)$ with $\boldsymbol{\Psi}^* \in \mathbb{R}^{q \times q}$ and $\boldsymbol{\Sigma}^* \in \mathbb{R}^{p \times p}$. Assume that $0 < C_1 \le \lambda_{\min}(\boldsymbol{\Sigma}^*) \le \lambda_{\max}(\boldsymbol{\Sigma}^*) \le 1/C_1 < \infty$ and $0 < C_2 \le \lambda_{\min}(\boldsymbol{\Psi}^*) \le \lambda_{\max}(\boldsymbol{\Psi}^*) \le 1/C_2 < \infty$ for some positive constants $C_1, C_2$. For any symmetric and positive definite matrix $\boldsymbol{\Omega} \in \mathbb{R}^{p \times p}$, we have

$$\max_{i,j}\left\{\frac{1}{np}\sum_{i=1}^n \mathbf{X}_i^\top \boldsymbol{\Omega}\mathbf{X}_i - \frac{1}{p}\mathbb{E}(\mathbf{X}^\top \boldsymbol{\Omega}\mathbf{X})\right\}_{i,j} = O_P\left(\sqrt{\frac{\log q}{np}}\right).$$

**Proof of Lemma B.1:** Consider a random matrix $\mathbf{X}$ following the matrix normal distribution such that $\text{vec}(\mathbf{X}) \sim \text{N}(\mathbf{0}; \boldsymbol{\Psi}^* \otimes \boldsymbol{\Sigma}^*)$. Let $\boldsymbol{\Lambda}^* = \boldsymbol{\Psi}^{*-1}$ and $\boldsymbol{\Omega}^* = \boldsymbol{\Sigma}^{*-1}$. Let $\mathbf{Y} := (\boldsymbol{\Omega}^*)^{1/2}\mathbf{X}(\boldsymbol{\Lambda}^*)^{1/2}$. According to the properties of matrix normal distribution [30], $\mathbf{Y}$ follows a matrix normal distribution such that $\text{vec}(\mathbf{Y}) \sim \text{N}(\mathbf{0}; \mathbb{1}_q \otimes \mathbb{1}_p)$, that is, all the entries of $\mathbf{Y}$ are i.i.d. standard Gaussian random variables. Next we rewrite the term $\mathbf{X}^\top \boldsymbol{\Omega} \mathbf{X}$ by $\mathbf{Y}$ and then simplify it. Simple algebra implies that

$$\mathbf{X}^\top \boldsymbol{\Omega} \mathbf{X} = (\boldsymbol{\Lambda}^*)^{-1/2}\mathbf{Y}^\top(\boldsymbol{\Omega}^*)^{-1/2}\boldsymbol{\Omega}(\boldsymbol{\Omega}^*)^{-1/2}\mathbf{Y}(\boldsymbol{\Lambda}^*)^{-1/2}.$$

When $\boldsymbol{\Omega}$ is symmetric and positive definite, the matrix $\mathbf{M} := (\boldsymbol{\Omega}^*)^{-1/2}\boldsymbol{\Omega}(\boldsymbol{\Omega}^*)^{-1/2} \in \mathbb{R}^{p \times p}$ is also symmetric and positive definite with Cholesky decomposition $\mathbf{U}^\top \mathbf{U}$, where $\mathbf{U} \in \mathbb{R}^{p \times p}$. Therefore,

$$\mathbf{X}^\top \boldsymbol{\Omega} \mathbf{X} = (\boldsymbol{\Lambda}^*)^{-1/2}\mathbf{Y}^\top\mathbf{U}^\top\mathbf{U}\mathbf{Y}(\boldsymbol{\Lambda}^*)^{-1/2}.$$

Moreover, denote the column of the matrix $(\boldsymbol{\Lambda}^*)^{-1/2}$ as $(\boldsymbol{\Lambda}^*)^{-1/2}_{(j)}$ and denote its row as $(\boldsymbol{\Lambda}^*)^{-1/2}_i$ for $i, j = 1, \ldots, q$. Define the standard basis $\mathbf{e}_i \in \mathbb{R}^q$ as the vector with 1 in its $i$-th entry and 0 in all the rest entries. The $(s, t)$-th entry of matrix $\mathbf{X}^\top \boldsymbol{\Omega} \mathbf{X}$ can be written as

$$\left\{\mathbf{X}^\top \boldsymbol{\Omega} \mathbf{X}\right\}_{s,t} = \mathbf{e}_s^\top \mathbf{X}^\top \boldsymbol{\Omega} \mathbf{X} \mathbf{e}_t = (\boldsymbol{\Lambda}^*)^{-1/2}_s \mathbf{Y}^\top \mathbf{U}^\top \mathbf{U} \mathbf{Y}(\boldsymbol{\Lambda}^*)^{-1/2}_{(t)}.$$

For the sample matrices $\mathbf{X}_1, \ldots, \mathbf{X}_n$, we apply similar transformation that $\mathbf{Y}_i = (\boldsymbol{\Omega}^*)^{1/2}\mathbf{X}_i(\boldsymbol{\Lambda}^*)^{1/2}$. We apply the above derivation to the sample-based quadratic term $\mathbf{X}_i^\top \boldsymbol{\Omega} \mathbf{X}_i$. Let $\mathbf{A} = (\mathbf{a}_1, \ldots, \mathbf{a}_n) \in \mathbb{R}^{p \times n}$ with $\mathbf{a}_i = \mathbf{U}\mathbf{Y}_i(\boldsymbol{\Lambda}^*)^{-1/2}_s \in \mathbb{R}^p$ and $\mathbf{B} = (\mathbf{b}_1, \ldots, \mathbf{b}_n) \in \mathbb{R}^{p \times n}$ with $\mathbf{b}_i = \mathbf{U}\mathbf{Y}_i(\boldsymbol{\Lambda}^*)^{-1/2}_t \in \mathbb{R}^p$. Then we have

$$
\begin{aligned}
\left\{\frac{1}{np}\sum_{i=1}^n \mathbf{X}_i^\top \boldsymbol{\Omega} \mathbf{X}_i\right\}_{s,t} &= \frac{1}{np}\sum_{i=1}^n \mathbf{a}_i^\top \mathbf{b}_i = \frac{1}{np}\sum_{i=1}^n\sum_{j=1}^p \mathbf{A}_{i,j}\mathbf{B}_{i,j} \\
&= \frac{1}{4np}\sum_{i=1}^n\sum_{j=1}^p \left\{(\mathbf{A}_{i,j} + \mathbf{B}_{i,j})^2 - (\mathbf{A}_{i,j} - \mathbf{B}_{i,j})^2\right\} \\
&= \frac{1}{4np}\left\{\|\text{vec}(\mathbf{A}) + \text{vec}(\mathbf{B})\|_2^2 + \|\text{vec}(\mathbf{A}) - \text{vec}(\mathbf{B})\|_2^2\right\}. \quad \text{(B.1)}
\end{aligned}
$$

Next we derive the explicit form of $\text{vec}(\mathbf{A})$ and $\text{vec}(\mathbf{B})$ in (B.1). Remind that $(\boldsymbol{\Lambda}^*)^{-1/2}_s$ is a vector of length $q$. By the property of matrix products, we can rewrite $\mathbf{a}_i = [(\boldsymbol{\Lambda}^*)^{-1/2}_s \otimes \mathbf{U}]\text{vec}(\mathbf{Y}_i)$, where $\otimes$ is the Kronecker product. Therefore, we have

$$
\begin{aligned}
\text{vec}(\mathbf{A}) &= \left[\mathbb{1}_n \otimes (\boldsymbol{\Lambda}^*)^{-1/2}_s \otimes \mathbf{U}\right]\mathbf{t} := \mathbf{Q}_1\mathbf{t}, \\
\text{vec}(\mathbf{B}) &= \left[\mathbb{1}_n \otimes (\boldsymbol{\Lambda}^*)^{-1/2}_t \otimes \mathbf{U}\right]\mathbf{t} := \mathbf{Q}_2\mathbf{t},
\end{aligned}
$$

where $\mathbf{t} = \left\{[\text{vec}(\mathbf{Y}_1)]^\top, \ldots, [\text{vec}(\mathbf{Y}_n)]^\top\right\}^\top \in \mathbb{R}^{npq}$ is a vector with $npq$ i.i.d. standard normal entries. Here $\mathbf{Q}_1 := \mathbb{1}_n \otimes (\boldsymbol{\Lambda}^*)^{-1/2}_s \otimes \mathbf{U}$ and $\mathbf{Q}_2 := \mathbb{1}_n \otimes (\boldsymbol{\Lambda}^*)^{-1/2}_t \otimes \mathbf{U}$ with $\mathbf{Q}_1, \mathbf{Q}_2 \in \mathbb{R}^{np \times npq}$. By the property of multivariate normal distribution, we have

$$
\begin{aligned}
\text{vec}(\mathbf{A}) + \text{vec}(\mathbf{B}) &\sim \text{N}\left(0; (\mathbf{Q}_1 + \mathbf{Q}_2)(\mathbf{Q}_1 + \mathbf{Q}_2)^\top\right) := \text{N}(0; \mathbf{H}_1), \\
\text{vec}(\mathbf{A}) - \text{vec}(\mathbf{B}) &\sim \text{N}\left(0; (\mathbf{Q}_1 - \mathbf{Q}_2)(\mathbf{Q}_1 - \mathbf{Q}_2)^\top\right) := \text{N}(0; \mathbf{H}_2).
\end{aligned}
$$

Next, we bound the spectral norm of two matrices $\mathbf{H}_1$ and $\mathbf{H}_2$. By the property of matrix norm and the fact that one matrix and its transpose matrix have the same spectral norm, we have

$$\|\mathbf{H}_1\|_2 \leq \|\mathbf{Q}_1\mathbf{Q}_1^\top\|_2 + 2\|\mathbf{Q}_1\mathbf{Q}_2^\top\|_2 + \|\mathbf{Q}_2\mathbf{Q}_2^\top\|_2,$$

then we bound each of these three terms individually. According to the definition of $\mathbf{Q}_1$ and the property of matrix Kronecker products, we have

$$
\begin{aligned}
\mathbf{Q}_1\mathbf{Q}_1^\top &= \left[\mathbb{1}_n \otimes (\boldsymbol{\Lambda}^*)^{-1/2}_s \otimes \mathbf{U}\right]\left[\mathbb{1}_n \otimes (\boldsymbol{\Lambda}^*)^{-1/2}_s \otimes \mathbf{U}\right]^\top \\
&= \mathbb{1}_n \otimes (\boldsymbol{\Lambda}^*)^{-1/2}_s[(\boldsymbol{\Lambda}^*)^{-1/2}_s]^\top \otimes \mathbf{M},
\end{aligned}
$$

where the last equality is due to the fact that $(\mathbf{C}_1 \otimes \mathbf{C}_2)^\top = \mathbf{C}_1^\top \otimes \mathbf{C}_2^\top$ and $(\mathbf{C}_1 \otimes \mathbf{C}_2)(\mathbf{C}_3 \otimes \mathbf{C}_4) = (\mathbf{C}_1\mathbf{C}_3) \otimes (\mathbf{C}_2\mathbf{C}_4)$ for any matrices $\mathbf{C}_1, \ldots, \mathbf{C}_4$ such that the matrix multiplications $\mathbf{C}_1\mathbf{C}_3$ and

$\mathbf{C}_2\mathbf{C}_4$ are valid. Moreover, we also use the Cholesky decomposition of $\mathbf{M}$, i.e., $\mathbf{M} = \mathbf{U}^\top\mathbf{U}$. Remind that $(\mathbf{\Lambda}^*)_s^{-1/2}[(\mathbf{\Lambda}^*)_s^{-1/2}]^\top \in \mathbb{R}$, therefore, the spectral norm $\mathbf{Q}_1\mathbf{Q}_1^\top$ can be written as

$$
\begin{aligned}
\|\mathbf{Q}_1\mathbf{Q}_1^\top\|_2 &= \left|(\mathbf{\Lambda}^*)_s^{-1/2}[(\mathbf{\Lambda}^*)_s^{-1/2}]^\top\right| \cdot \|\mathbb{1}_n\|_2\|\mathbf{M}\|_2 \\
&\leq \|\mathbf{\Psi}^*\|_2\|\mathbf{M}\|_2 \leq (1+\alpha/C_1)/C_2.
\end{aligned}
$$

Here the first inequality is because $\|\mathbb{1}_n\|_2 = 1$ and

$$
\left|(\mathbf{\Lambda}^*)_s^{-1/2}[(\mathbf{\Lambda}^*)_s^{-1/2}]^\top\right| = \left\|[(\mathbf{\Lambda}^*)_s^{-1/2}]^\top(\mathbf{\Lambda}^*)_s^{-1/2}\right\|_2 \leq \max_j\left\|[(\mathbf{\Psi}^*)_j^{1/2}]^\top(\mathbf{\Psi}^*)_j^{1/2}\right\|_2
$$

$$
\leq \left\|\sum_{j=1}^q[(\mathbf{\Psi}^*)_j^{1/2}]^\top(\mathbf{\Psi}^*)_j^{1/2}\right\|_2 = \|\mathbf{\Psi}^*\|_2,
$$

and the second inequality is because $\|\mathbf{\Psi}^*\|_2 \leq 1/C_2$ and

$$
\begin{aligned}
\|\mathbf{M}\|_2 &= \left\|(\mathbf{\Omega}^*)^{-1/2}\mathbf{\Omega}(\mathbf{\Omega}^*)^{-1/2}\right\|_2 = \|(\mathbf{\Omega}^*)^{-1/2}(\mathbf{\Omega}-\mathbf{\Omega}^*)(\mathbf{\Omega}^*)^{-1/2} + \mathbb{1}_p\|_2 \\
&\leq \|(\mathbf{\Omega}^*)^{-1/2}\|_2^2\|\mathbf{\Omega}-\mathbf{\Omega}^*\|_2 + 1 \leq \|\mathbf{\Sigma}^*\|_2\|\mathbf{\Omega}-\mathbf{\Omega}^*\|_F + 1 \leq 1 + \alpha/C_1.
\end{aligned}
$$

Similarly, we have $\|\mathbf{Q}_2\mathbf{Q}_2^\top\|_2 \leq (1+\alpha/C_1)/C_2$. For $\|\mathbf{Q}_1\mathbf{Q}_2^\top\|_2$, similar arguments imply that

$$
\mathbf{Q}_1\mathbf{Q}_2^\top = \mathbb{1}_n\otimes(\mathbf{\Lambda}^*)_s^{-1/2}[(\mathbf{\Lambda}^*)_t^{-1/2}]^\top\otimes\mathbf{M},
$$

and hence its spectral norm is bounded as

$$
\begin{aligned}
\|\mathbf{Q}_1\mathbf{Q}_2^\top\|_2 &= \left|(\mathbf{\Lambda}^*)_s^{-1/2}[(\mathbf{\Lambda}^*)_t^{-1/2}]^\top\right| \cdot \|\mathbb{1}_n\|_2\|\mathbf{M}\|_2 \\
&\leq \|\mathbf{\Psi}^*\|_2\|\mathbf{M}\|_2 \leq (1+\alpha/C_1)/C_2,
\end{aligned}
$$

where the first inequality is because the above derivation and the Cauchy-Schwarz inequality. Specifically, let $\mathbf{\Psi}^* = (\mathbf{\Psi}_{i,j}^*)$, we have

$$
\left|(\mathbf{\Lambda}^*)_s^{-1/2}[(\mathbf{\Lambda}^*)_t^{-1/2}]^\top\right| = \sqrt{(\mathbf{\Psi}^*)_s[(\mathbf{\Psi}^*)_t]^\top} = \Big[\sum_{j=1}^q\mathbf{\Psi}_{s,j}^*\mathbf{\Psi}_{t,j}^*\Big]^{1/2}
$$

$$
\leq \left\{(\sum_{j=1}^q\mathbf{\Psi}_{s,j}^{*2})(\sum_{j=1}^q\mathbf{\Psi}_{t,j}^{*2})\right\}^{1/4} \leq \sqrt{\|\mathbf{\Psi}^*\|_2\|\mathbf{\Psi}^*\|_2} \leq C_2^{-1}.
$$

Applying the same techniques to $\|\mathbf{H}_2\|_2$, we have

$$
\begin{aligned}
\|\mathbf{H}_1\|_2 &\leq 4(1+\alpha/C_1)/C_2, &\text{(B.2)} \\
\|\mathbf{H}_2\|_2 &\leq 4(1+\alpha/C_1)/C_2. &\text{(B.3)}
\end{aligned}
$$

Next, we apply Lemma C.3 to bound the $(s,t)$-th entry of the differential matrix between the sample-based term and its expectation. Denote $\rho_{s,t} := [p^{-1}\mathbb{E}(\mathbf{X}^\top\mathbf{\Omega}\mathbf{X})]_{s,t}$. According to the derivation in (B.1), we have

$$
\begin{aligned}
&\left\{\frac{1}{np}\sum_{i=1}^n\mathbf{X}_i^\top\mathbf{\Omega}\mathbf{X}_i - \frac{1}{p}\mathbb{E}(\mathbf{X}^\top\mathbf{\Omega}\mathbf{X})\right\}_{s,t} \\
&= \left[\frac{1}{4np}\sum_{i,j}(a_{ij}+b_{ij})^2 - \frac{\mathbf{\Delta}_{s,t}+\rho_{s,t}}{2}\right] - \left[\frac{1}{4np}\sum_{i,j}(a_{ij}-b_{ij})^2 - \frac{\mathbf{\Delta}_{s,t}-\rho_{s,t}}{2}\right], \text{(B.4)}
\end{aligned}
$$

where $\mathbf{\Delta}_{s,t}$ is defined as

$$
\mathbf{\Delta}_{s,t} := \mathbb{E}\Big\{(4np)^{-1}\sum_{i,j}[(a_{ij}+b_{ij})^2 + (a_{ij}-b_{ij})^2]\Big\}.
$$

Furthermore, according to the definition of $\rho_{s,t}$ and (B.1), we have $\mathbb{E}\{(4np)^{-1}\sum_{i=1}^n\sum_{j=1}^p[(a_{ij}+b_{ij})^2 - (a_{ij}-b_{ij})^2]\} = \rho_{s,t}$. Therefore, we have

$$
\mathbb{E}\{(4np)^{-1}\sum_{i,j}(a_{ij}+b_{ij})^2\} = \frac{\mathbf{\Delta}_{s,t}+\rho_{s,t}}{2}, \tag{B.5}
$$

$$
\mathbb{E}\{(4np)^{-1}\sum_{i,j}(a_{ij}-b_{ij})^2\} = \frac{\mathbf{\Delta}_{s,t}-\rho_{s,t}}{2}. \tag{B.6}
$$

Therefore, (B.4) implies that, for any $\delta > 0$,

$$\mathbb{P}\Big[\Big|\Big\{\frac{1}{np}\sum_{i=1}^{n}\mathbf{X}_i^{\top}\mathbf{\Omega}\mathbf{X}_i - \frac{1}{p}\mathbb{E}(\mathbf{X}^{\top}\mathbf{\Omega}\mathbf{X})\Big\}_{s,t}\Big| \geq \delta\Big] \leq$$

$$\underbrace{\mathbb{P}\Big[\Big|\frac{1}{np}\sum_{i,j}(a_{ij}+b_{ij})^2 - 2(\mathbf{\Delta}_{s,t}+\rho_{s,t})\Big| > 2\delta\Big]}_{I_1} + \underbrace{\mathbb{P}\Big[\Big|\frac{1}{np}\sum_{i,j}(a_{ij}-b_{ij})^2 - 2(\mathbf{\Delta}_{s,t}-\rho_{s,t})\Big| > 2\delta\Big]}_{I_2}.$$

Remind that $\sum_{i=1}^{n}\sum_{j=1}^{p}(a_{ij}+b_{ij})^2 = \mathrm{vec}(\mathbf{A}) + \mathrm{vec}(\mathbf{B}) \sim \mathrm{N}(0; \mathbf{H}_1)$ and $\sum_{i=1}^{n}\sum_{j=1}^{p}(a_{ij}-b_{ij})^2 = \mathrm{vec}(\mathbf{A}) - \mathrm{vec}(\mathbf{B}) \sim \mathrm{N}(0; \mathbf{H}_2)$. According to (B.5) and (B.6), we apply Lemma C.3 to obtain

$$I_1 \leq 2\exp\Big\{-\frac{np}{2}\Big(\frac{\delta}{2\|\mathbf{H}_1\|_2} - \frac{2}{\sqrt{np}}\Big)^2\Big\} + 2\exp(-np/2),$$

$$I_2 \leq 2\exp\Big\{-\frac{np}{2}\Big(\frac{\delta}{2\|\mathbf{H}_2\|_2} - \frac{2}{\sqrt{np}}\Big)^2\Big\} + 2\exp(-np/2).$$

Finally, in order to derive the convergence rate of the maximal difference over all index $(s,t)$, we employ the max sum inequality. That is, for random variables $x_1,\ldots,x_n$, we have $\mathbb{P}(\max_i x_i \geq t) \leq \sum_{i=1}^{n}\mathbb{P}(x_i \geq t) \leq n\max_i\mathbb{P}(x_i \geq t)$. This together with (B.2) and (B.3) imply that

$$\mathbb{P}\Big[\max_{(s,t)}\Big\{\frac{1}{np}\sum_{i=1}^{n}\mathbf{X}_i^{\top}\mathbf{\Omega}\mathbf{X}_i - \frac{1}{p}\mathbb{E}(\mathbf{X}^{\top}\mathbf{\Omega}\mathbf{X})\Big\}_{s,t} \geq \delta\Big]$$

$$\leq 4q^2\exp\Big\{-\frac{np}{2}\Big[\frac{\delta C_1 C_2}{8(C_1+\alpha)} - \frac{2}{\sqrt{np}}\Big]^2\Big\} + 4q^2\exp(-np/2). \qquad (B.7)$$

Let $\delta = 8(C_1+\alpha)(C_1C_2)^{-1}[4\sqrt{\log q/(np)} + 3(np)^{-1/2}]$ in (B.7) which satisfies the condition in Lemma C.3 since $\delta > 2(np)^{-1/2}$ when $q$ is sufficiently large. Therefore, we obtain the desirable conclusion that, with high probability,

$$\max_{(s,t)}\Big\{\frac{1}{np}\sum_{i=1}^{n}\mathbf{X}_i^{\top}\mathbf{\Omega}\mathbf{X}_i - \frac{1}{p}\mathbb{E}(\mathbf{X}^{\top}\mathbf{\Omega}\mathbf{X})\Big\}_{s,t} = O_P\Big(\sqrt{\frac{\log q}{np}}\Big).$$

This ends the proof of Lemma B.1. ∎

**Lemma B.2.** Assume i.i.d. tensor data $\mathcal{T}, \mathcal{T}_1,\ldots,\mathcal{T}_n \in \mathbb{R}^{m_1\times m_2\times\cdots\times m_K}$ follows the tensor normal distribution $\mathrm{TN}(\mathbf{0}; \mathbf{\Sigma}_1^*,\ldots,\mathbf{\Sigma}_K^*)$. Assume Condition 3.2 holds. For any symmetric and positive definite matrices $\mathbf{\Omega}_j \in \mathbb{R}^{m_j\times m_j}, j \neq k$, we have

$$\mathbb{E}[\mathbf{S}_k] = \frac{m_k[\prod_{j\neq k}\mathrm{tr}(\mathbf{\Sigma}_j^*\mathbf{\Omega}_j)]}{m}\mathbf{\Sigma}_k^*,$$

for $\mathbf{S}_k = \frac{m_k}{nm}\sum_{i=1}^{n}\mathbf{V}_i\mathbf{V}_i^{\top}$ with $\mathbf{V}_i = \big[\mathcal{T}_i \times \{\mathbf{\Omega}_1^{1/2},\ldots,\mathbf{\Omega}_{k-1}^{1/2},\mathbb{1}_{m_k},\mathbf{\Omega}_{k+1}^{1/2},\ldots,\mathbf{\Omega}_K^{1/2}\}\big]_{(k)}$ and $m = \prod_{k=1}^{K}m_k$. Moreover, we have

$$\max_{s,t}\Big\{\mathbf{S}_k - \frac{m_k[\prod_{j\neq k}\mathrm{tr}(\mathbf{\Sigma}_j^*\mathbf{\Omega}_j)]}{m}\mathbf{\Sigma}_k^*\Big\}_{s,t} = O_P\Big(\sqrt{\frac{m_k\log m_k}{nm}}\Big). \qquad (B.8)$$

**Proof of Lemma B.2:** The proof follows by carefully examining the distribution of $\mathbf{V}_i$ and then applying Lemma B.1. We only show the case with $K = 3$ and $k = 1$. The extension to a general $K$ follows similarly.

According to the property of mode-$k$ tensor multiplication, we have $\mathbf{V}_i = [\mathcal{T}_i]_{(1)}(\mathbf{\Omega}_3^{1/2}\otimes\mathbf{\Omega}_2^{1/2})$, and hence

$$\mathbf{S}_1 = \frac{1}{nm_2m_3}\sum_{i=1}^{n}[\mathcal{T}_i]_{(1)}(\mathbf{\Omega}_3^{1/2}\otimes\mathbf{\Omega}_2^{1/2})(\mathbf{\Omega}_3^{1/2}\otimes\mathbf{\Omega}_2^{1/2})[\mathcal{T}_i]_{(1)}^{\top}$$

$$= \frac{1}{nm_2m_3}\sum_{i=1}^{n}[\mathcal{T}_i]_{(1)}(\mathbf{\Omega}_3\otimes\mathbf{\Omega}_2)[\mathcal{T}_i]_{(1)}^{\top}.$$

When tensor $\mathcal{T}_i \sim \text{TN}(\mathbf{0}; \boldsymbol{\Sigma}_1^*, \boldsymbol{\Sigma}_2^*, \boldsymbol{\Sigma}_3^*)$, the property of mode-$k$ tensor multiplication shown in Proposition 2.1 in [31] implies that

$$[\mathcal{T}_i]_{(1)} \in \mathbb{R}^{m_1 \times (m_2 m_3)} \sim \text{MN}(\mathbf{0}; \boldsymbol{\Sigma}_1^*, \boldsymbol{\Sigma}_3^* \otimes \boldsymbol{\Sigma}_2^*),$$

where $\text{MN}(\mathbf{0}; \boldsymbol{\Sigma}_1^*, \boldsymbol{\Sigma}_3^* \otimes \boldsymbol{\Sigma}_2^*)$ is the matrix-variate normal [32] such that the row covariance matrix of $[\mathcal{T}_i]_{(1)}$ is $\boldsymbol{\Sigma}_1^*$ and the column covariance matrix of $[\mathcal{T}_i]_{(1)}$ is $\boldsymbol{\Sigma}_3^* \otimes \boldsymbol{\Sigma}_2^*$. Therefore, in order to show (B.8), according to Lemma B.1, it is sufficient to show

$$\mathbb{E}[\mathbf{S}_1] = \frac{\text{tr}(\boldsymbol{\Sigma}_3^* \boldsymbol{\Omega}_3)\text{tr}(\boldsymbol{\Sigma}_2^* \boldsymbol{\Omega}_2)}{m_2 m_3} \boldsymbol{\Sigma}_1^*. \tag{B.9}$$

According to the distribution of $[\mathcal{T}_i]_{(1)}$, we have

$$\mathbf{V}_i \sim \text{MN}\left(\mathbf{0}; \boldsymbol{\Sigma}_1^*, (\boldsymbol{\Omega}_3^{1/2} \otimes \boldsymbol{\Omega}_2^{1/2})(\boldsymbol{\Sigma}_3^* \otimes \boldsymbol{\Sigma}_2^*)(\boldsymbol{\Omega}_3^{1/2} \otimes \boldsymbol{\Omega}_2^{1/2})\right),$$

and hence

$$\mathbf{V}_i^\top \sim \text{MN}\left(\mathbf{0}; (\boldsymbol{\Omega}_3^{1/2} \otimes \boldsymbol{\Omega}_2^{1/2})(\boldsymbol{\Sigma}_3^* \otimes \boldsymbol{\Sigma}_2^*)(\boldsymbol{\Omega}_3^{1/2} \otimes \boldsymbol{\Omega}_2^{1/2}), \boldsymbol{\Sigma}_1^*\right).$$

Therefore, according to Lemma C.1, we have

$$\mathbb{E}[\mathbf{V}_i \mathbf{V}_i^\top] = \boldsymbol{\Sigma}_1^* \text{tr}\left[(\boldsymbol{\Omega}_3 \otimes \boldsymbol{\Omega}_2)(\boldsymbol{\Sigma}_3^* \otimes \boldsymbol{\Sigma}_2^*)\right] = \boldsymbol{\Sigma}_1^* \text{tr}(\boldsymbol{\Sigma}_3^* \boldsymbol{\Omega}_3)\text{tr}(\boldsymbol{\Sigma}_2^* \boldsymbol{\Omega}_2),$$

which implies (B.9) according to the definition of $\mathbf{S}_1$. Finally, applying Lemma B.1 to $\mathbf{S}_1$ leads to the desirable result. This ends the proof of Lemma B.2. ∎

The following lemma establishes the rate of convergence of the sample covariance matrix in max norm.

**Lemma B.3.** Assume i.i.d. tensor data $\mathcal{T}, \mathcal{T}_1, \ldots, \mathcal{T}_n \in \mathbb{R}^{m_1 \times m_2 \times \cdots \times m_K}$ follows the tensor normal distribution $\text{TN}(\mathbf{0}; \boldsymbol{\Sigma}_1^*, \cdots, \boldsymbol{\Sigma}_K^*)$, and assume Condition 3.2 holds. Let $\widehat{\boldsymbol{\Omega}}_j \in \mathbb{R}^{m_j \times m_j}, j \neq k$, be the estimated precision matrix from Algorithm 1 with iteration number $T = 1$. Denote the $k$-th sample covariance matrix as

$$\widehat{\mathbf{S}}_k = \frac{m_k}{nm} \sum_{i=1}^n \widehat{\mathbf{V}}_i \widehat{\mathbf{V}}_i^\top,$$

with $m = \prod_{k=1}^K m_k$ and $\widehat{\mathbf{V}}_i := \left[\mathcal{T}_i \times \{\widehat{\boldsymbol{\Omega}}_1^{1/2}, \ldots, \widehat{\boldsymbol{\Omega}}_{k-1}^{1/2}, \mathbb{1}_{m_k}, \widehat{\boldsymbol{\Omega}}_{k+1}^{1/2}, \ldots, \widehat{\boldsymbol{\Omega}}_K^{1/2}\}\right]_{(k)}$. We have

$$\max_{s,t} \left[\widehat{\mathbf{S}}_k - \boldsymbol{\Sigma}_k^*\right]_{s,t} = O_P\left(\max_{j=1,\ldots,K} \sqrt{\frac{(m_j + s_j)\log m_j}{nm}}\right). \tag{B.10}$$

**Proof of Lemma B.3:** The proof follows by decomposing the $\widehat{\mathbf{S}}_k - \boldsymbol{\Sigma}_k^*$ into two parts and then applying Lemma B.2 and Theorem 3.5 for each part to bound the final error.

Note that the triangle inequality implies that

$$\left\|\widehat{\mathbf{S}}_k - \boldsymbol{\Sigma}_k^*\right\|_\infty \leq \underbrace{\left\|\widehat{\mathbf{S}}_k - \frac{m_k[\prod_{j\neq k}\text{tr}(\boldsymbol{\Sigma}_j^* \widehat{\boldsymbol{\Omega}}_j)]}{m}\boldsymbol{\Sigma}_k^*\right\|_\infty}_{I_1} + \underbrace{\left\|\frac{m_k[\prod_{j\neq k}\text{tr}(\boldsymbol{\Sigma}_j^* \widehat{\boldsymbol{\Omega}}_j)]}{m}\boldsymbol{\Sigma}_k^* - \boldsymbol{\Sigma}_k^*\right\|_\infty}_{I_2}.$$

Note that here the covariance matrix $\widehat{\mathbf{S}}_k$ is constructed based on the estimators $\widehat{\boldsymbol{\Omega}}_j, j \neq k$. According to (B.8) in Lemma B.2, we have

$$I_1 = O_P\left(\sqrt{\frac{m_k \log m_k}{nm}}\right).$$

The remainder part is to bound the error $I_2$. Note that $\text{tr}(\boldsymbol{\Sigma}_j^* \boldsymbol{\Omega}_j^*) = \text{tr}(\mathbb{1}_{m_j}) = m_j$. Therefore,

$$I_2 = \underbrace{\left|\frac{m_k}{m}\left[\prod_{j\neq k}\text{tr}(\boldsymbol{\Sigma}_j^* \widehat{\boldsymbol{\Omega}}_j) - \prod_{j\neq k}\text{tr}(\boldsymbol{\Sigma}_j^* \boldsymbol{\Omega}_j^*)\right]\right|}_{I_3} \|\boldsymbol{\Sigma}_k^*\|_\infty.$$

Given that $\|\mathbf{\Sigma}_k^*\|_\infty = O_P(1)$, it is sufficient to bound the coefficient $I_3$. We only demonstrate the proofs with $K = 3$ and $k = 1$. The extension to a general $K$ follows similarly. In this case, we have

$$
\begin{aligned}
I_3 &= \frac{m_1}{m}\left|\mathrm{tr}(\mathbf{\Sigma}_2^*\widehat{\mathbf{\Omega}}_2)\mathrm{tr}(\mathbf{\Sigma}_3^*\widehat{\mathbf{\Omega}}_3) - \mathrm{tr}(\mathbf{\Sigma}_2^*\mathbf{\Omega}_2^*)\mathrm{tr}(\mathbf{\Sigma}_3^*\mathbf{\Omega}_3^*)\right| \\
&\leq \left|\frac{\mathrm{tr}(\mathbf{\Sigma}_2^*\widehat{\mathbf{\Omega}}_2)\mathrm{tr}[\mathbf{\Sigma}_3^*(\widehat{\mathbf{\Omega}}_3 - \mathbf{\Omega}_3^*)]}{m_2 m_3}\right| + \left|\frac{\mathrm{tr}[\mathbf{\Sigma}_2^*(\widehat{\mathbf{\Omega}}_2 - \mathbf{\Omega}_2^*)]\mathrm{tr}(\mathbf{\Sigma}_3^*\mathbf{\Omega}_3^*)}{m_2 m_3}\right|.
\end{aligned}
$$

According to the proof of Theorem 3.5, we have $C_1 \leq \mathrm{tr}(\mathbf{\Sigma}_j^*\mathbf{\Omega}_j)/m_j \leq 1/C_1$ for any $j = 1, \ldots, K$ and some constant $C_1 > 0$. Moreover, we have $\mathrm{tr}(\mathbf{\Sigma}_3^*\mathbf{\Omega}_3^*) = m_3$. Therefore, we have

$$
I_3 \leq \left|\frac{\mathrm{tr}[\mathbf{\Sigma}_3^*(\widehat{\mathbf{\Omega}}_3 - \mathbf{\Omega}_3^*)]}{m_3}\right| + \left|\frac{\mathrm{tr}[\mathbf{\Sigma}_2^*(\widehat{\mathbf{\Omega}}_2 - \mathbf{\Omega}_2^*)]}{m_2}\right|.
$$

Here $\mathrm{tr}[\mathbf{\Sigma}_j^*(\widehat{\mathbf{\Omega}}_j - \mathbf{\Omega}_j^*)] \leq \|\mathbf{\Sigma}_j^*\|_F\|\widehat{\mathbf{\Omega}}_j - \mathbf{\Omega}_j^*\|_F \leq \sqrt{m_j}\|\mathbf{\Sigma}_j^*\|_2\|\widehat{\mathbf{\Omega}}_j - \mathbf{\Omega}_j^*\|_F$. According to Condition 3.2, $\|\mathbf{\Sigma}_j^*\|_2 = O_P(1)$. This together with Theorem 3.5 implies that

$$
I_3 = O_P\left(\sqrt{\frac{(m_3 + s_3)\log m_3}{nm}} + \sqrt{\frac{(m_2 + s_2)\log m_2}{nm}}\right).
$$

By generalizing it to a general $K$ and $k$, we have that

$$
I_3 = O_P\left(\max_{j \neq k}\sqrt{\frac{(m_j + s_j)\log m_j}{nm}}\right),
$$

and hence

$$
\left\|\widehat{\mathbf{S}}_k - \mathbf{\Sigma}_k^*\right\|_\infty = O_P\left(\sqrt{\frac{m_k \log m_k}{nm}} + \max_{j \neq k}\sqrt{\frac{(m_j + s_j)\log m_j}{nm}}\right),
$$

which leads to the desirable result. This ends the proof of Lemma B.3. ∎

## C  Auxiliary lemmas

**Lemma C.1.** Assume a random matrix $\mathbf{X} \in \mathbb{R}^{p \times q}$ follows the matrix-variate normal distribution such that $\mathrm{vec}(\mathbf{X}) \sim \mathrm{N}(\mathbf{0}; \mathbf{\Psi}^* \otimes \mathbf{\Sigma}^*)$ with $\mathbf{\Psi}^* \in \mathbb{R}^{q \times q}$ and $\mathbf{\Sigma}^* \in \mathbb{R}^{p \times p}$. Then for any symmetric and positive definite matrix $\mathbf{\Omega} \in \mathbb{R}^{p \times p}$, we have $\mathbb{E}(\mathbf{X}^\top \mathbf{\Omega} \mathbf{X}) = \mathbf{\Psi}^*\mathrm{tr}(\mathbf{\Omega}\mathbf{\Sigma}^*)$.

**Proof of Lemma C.1:** Since the matrix $\mathbf{\Omega}$ is symmetric and positive definite, it has the Cholesky decomposition $\mathbf{\Omega} = \mathbf{V}^\top \mathbf{V}$, where $\mathbf{V}$ is upper triangular with positive diagonal entries. Let $\mathbf{Y} := \mathbf{V}\mathbf{X}$ and denote the $j$-th row of matrix $\mathbf{Y}$ as $\mathbf{y}_j = (y_{j,1}, \ldots, y_{j,q})$. We have $\mathbb{E}(\mathbf{X}^\top \mathbf{\Omega} \mathbf{X}) = \mathbb{E}(\mathbf{Y}^\top \mathbf{Y}) = \sum_{j=1}^p \mathbb{E}(\mathbf{y}_j^\top \mathbf{y}_j)$. Here $\mathbf{y}_j = \mathbf{v}_j \mathbf{X}$ with $\mathbf{v}_j$ the $j$-th row of $\mathbf{V}$. Denote the $i$-th column of matrix $\mathbf{X}$ as $\mathbf{x}_{(i)}$, we have $y_{j,i} = \mathbf{v}_j \mathbf{x}_{(i)}$. Therefore, the $(s,t)$-th entry of $\mathbb{E}(\mathbf{y}_j^\top \mathbf{y}_j)$ is

$$
\left[\mathbb{E}(\mathbf{y}_j^\top \mathbf{y}_j)\right]_{(s,t)} = \mathbb{E}[\mathbf{v}_j \mathbf{x}_{(s)} \mathbf{v}_j \mathbf{x}_{(t)}] = \mathbf{v}_j \mathbb{E}[\mathbf{x}_{(s)} \mathbf{x}_{(t)}^\top]\mathbf{v}_j^\top = \mathbf{v}_j \mathbf{\Psi}_{s,t}^* \mathbf{\Sigma}^* \mathbf{v}_j^\top,
$$

where $\mathbf{\Psi}_{s,t}^*$ is the $(s,t)$-th entry of $\mathbf{\Psi}^*$. The last equality is due to $\mathrm{vec}(\mathbf{X}) = (\mathbf{x}_{(1)}^\top, \ldots, \mathbf{x}_{(q)}^\top)^\top \sim \mathrm{N}(\mathbf{0}; \mathbf{\Psi}^* \otimes \mathbf{\Sigma}^*)$ Therefore, we have

$$
\mathbb{E}(\mathbf{X}^\top \mathbf{\Omega} \mathbf{X}) = \sum_{j=1}^p \mathbb{E}(\mathbf{y}_j^\top \mathbf{y}_j) = \mathbf{\Psi}^* \sum_{j=1}^p \mathbf{v}_j \mathbf{\Sigma}^* \mathbf{v}_j^\top = \mathbf{\Psi}^* \mathrm{tr}\left(\sum_{j=1}^p \mathbf{v}_j^\top \mathbf{v}_j \mathbf{\Sigma}^*\right) = \mathbf{\Psi}^* \mathrm{tr}(\mathbf{\Omega}\mathbf{\Sigma}^*).
$$

This ends the proof of Lemma C.1. ∎

The following lemma is stated by [24].

**Lemma C.2.** Let random variables $x_1, \ldots, x_n \in \mathbb{R}$ be i.i.d. drawn from standard normal $N(0; 1)$ and denote $\mathbf{x} = (x_1, \ldots, x_n)^\top \in \mathbb{R}^n$ be a random vector. For a function $f : \mathbb{R}^n \to \mathbb{R}$ with Lipschitz constant $L$, that is, for any vectors $\mathbf{v}_1, \mathbf{v}_2 \in \mathbb{R}^n$, there exists $L \geq 0$ such that $|f(\mathbf{v}_1) - f(\mathbf{v}_2)| \leq L\|\mathbf{v}_1 - \mathbf{v}_2\|_2$. Then, for any $t > 0$, we have

$$\mathbb{P}\left\{|f(\mathbf{x}) - \mathbb{E}[f(\mathbf{x})]| > t\right\} \leq 2\exp\left(-\frac{t^2}{2L^2}\right).$$

The following lemma is useful for the proof of Lemma B.1. A similar statement was given in Lemma I.2 of [33].

**Lemma C.3.** Suppose that a $d$-dimensional Gaussian random vector $\mathbf{y} \sim N(0; \mathbf{Q})$, Then, for any $t > 2/\sqrt{d}$, we have

$$\mathbb{P}\left[\frac{1}{d}\big|\|\mathbf{y}\|_2^2 - \mathbb{E}(\|\mathbf{y}\|_2^2)\big| > 4t\|\mathbf{Q}\|_2\right] \leq 2\exp\left\{-\frac{d(t - 2/\sqrt{d})^2}{2}\right\} + 2\exp(-d/2).$$

**Proof of Lemma C.3:** Note that $\mathbb{E}(\|\mathbf{y}\|_2^2) \leq [\mathbb{E}(\|\mathbf{y}\|_2)]^2$ and hence

$$\|\mathbf{y}\|_2^2 - \mathbb{E}(\|\mathbf{y}\|_2^2) \leq [\|\mathbf{y}\|_2 - \mathbb{E}(\|\mathbf{y}\|_2)][\|\mathbf{y}\|_2 + \mathbb{E}(\|\mathbf{y}\|_2)].$$

The term $(\|\mathbf{y}\|_2 - \mathbb{E}(\|\mathbf{y}\|_2))$ can be bounded via the concentration inequality in Lemma C.2 by noting that $\|\mathbf{y}\|_2$ is a Lipschitz function of Gaussian random vector $\mathbf{y}$. The term $\|\mathbf{y}\|_2 + \mathbb{E}(\|\mathbf{y}\|_2)$ can also be bounded by the large deviation bound since $\mathbf{y}$ is a Gaussian random vector. This ends the proof of Lemma C.3. ∎

# D    Additional simulation results

In this section, we explain the details in generating the true precision matrices and then show additional numerical results.

**Triangle:** For each $k = 1, \ldots, K$, we construct the covariance matrix $\mathbf{\Sigma}_k \in \mathbb{R}^{m_k \times m_k}$ such that its $(i, j)$-th entry is $[\mathbf{\Sigma}_k]_{i,j} = \exp(-|h_i - h_j|/2)$ with $h_1 < h_2 < \cdots < h_{m_k}$. The difference $h_i - h_{i-1}$ with $i = 2, \ldots, m_k$ is generated independently and identically from $\text{Unif}(0.5, 1)$. This generated covariance matrix mimics the autoregressive process of order one, i.e., AR(1). We set $\mathbf{\Omega}_k^* = \mathbf{\Sigma}_k^{-1}$.

**Nearest neighbor:** For each $k = 1, \ldots, K$, we construct the precision matrix $\mathbf{\Omega}_k \in \mathbb{R}^{m_k \times m_k}$ directly from a four nearest-neighbor network. We first randomly pick $m_k$ points from a unit square and compute all pairwise distances among the points. We then search for the four nearest-neighbors of each point and a pair of symmetric entries in the precision matrix $\mathbf{\Omega}_k$ that has a random chosen value from $[-1, -0.5] \cup [0.5, 1]$. To ensure its positive definite property, we let the final precision matrix as $\mathbf{\Omega}_k^* = \mathbf{\Omega}_k + (|\lambda_{\min}(\mathbf{\Omega}_k) + 0.2| \cdot \mathbb{1}_{m_k})$, where $\lambda_{\min(\cdot)}$ refers to the smallest eigenvalue.

The additional error criterions for comparison are the averaged estimation errors in Frobinusm norm and max norm, i.e.,

$$\frac{1}{K}\sum_{k=1}^{K}\|\widehat{\mathbf{\Omega}}_k - \mathbf{\Omega}_k^*\|_F, \quad \frac{1}{K}\sum_{k=1}^{K}\|\widehat{\mathbf{\Omega}}_k - \mathbf{\Omega}_k^*\|_\infty.$$

Note that these two criterions are only available to the P-MLE method and our Tlasso. The direct Glasso method estimate the whole Kronecker product and hence could not produce the estimator for each precision matrix.

Remind that, as we show in Theorem 3.5 and Theorem 3.9, the estimation error for the $k$-th precision matrix is $O_p(\sqrt{m_k(m_k + s_k)\log m_k/(nm)})$ in Frobenius norm or $O_p(\sqrt{m_k \log m_k/(nm)})$ in max norm, where $m = m_1 m_2 m_3$ in this example. These theoretical findings are supported by the numerical results in Figure 2. In particular, as sample size $n$ increases from Scenario s1 to s2, the estimation errors in both Frobenius norm and max norm expectedly decrease. From Scenario s1 to s3, one dimension $m_1$ increases from 10 to 100, and other dimensions $m_2, m_3$ decrease from 10 to 5, in which case the averaged estimation error in max norm is decreasing, while the error in Frobenius norm increases due to its additional $\sqrt{m_k + s_k}$ effect. Moreover, compared to the P-MLE

Figure 2: Averaged estimation errors of the precision matrices in Frobenius norm and max norm of each method in Simulations 1&2, respectively. The left two plots are for Simulation 1, and the right two are for Simulation 2.

method, our Tlasso is better in Scenarios s1 and s2 and is worse in Scenario s3 in Frobenius norm. However, in terms of the max norm, our Talsso delivers significant better performance in 4 scenarios and comparable results in the rest 2 scenarios.