[Reviews · NeurIPS 2015]

Submitted by Assigned_Reviewer_1

In this paper, the authors study the estimation of sparse graphical models, which is formulated as a nonconvex optimization problem. An alternating minimization algorithm is proposed to solve it. The optimal statistical rate of convergence and the consistent graph recovery properties are discussed. The paper is organized very well and presented clearly. Only few questions:

(1) The sparsity of the solution \Omega. Problem (2.2) is formulated to encourage the sparsity of each precision matrix \Omega_i. In simulation part, there is no sparsity result. Page 8, lines 414-415, "Tlasso tends to include more non-connected edges than other methods", does this mean the computational results of Tlasso are not quite sparse? Please give an explanation.

(2) Page 4, line 206-210. The initialization of Algorithm 1. You claim the obtained estimators are insensitive to the choice of the initialization. In your simulations, set 1_mk as the initialization got better numerical performance. Why does this happen? How worse can it be for the random initialization? Can you give a short discussion?
Summary: In this paper, the authors study the estimation of sparse graphical models, which is formulated as a nonconvex optimization problem. An alternating minimization algorithm is proposed to solve it. The optimal statistical rate of convergence and the consistent graph recovery properties are discussed. The paper is organized very well and presented clearly.

Submitted by Assigned_Reviewer_2

after rebuttal:

As the irrelevance of the initialization is one of the main contributions of this paper, it is strange to me that the authors adopted the same initialization as previous papers. This really raises a big concern on the work.

============ In my opinion, this is a technically sound paper with nontrivial results. Following a recent thread of research on alternating minimization applied to different statistical problems, the authors established interesting results on recovering precision matrices from tensor normal distributions, and hence can be viewed as another manifestation of the power of alternating minimization to statistical problems. Theorem 3.1, though the simplest result in this paper, presents the core idea and hence is perhaps the most important observation in this paper for followup researches. The proofs of other theorems are more technical, but are clear enough for me to follow, and contain no unfixable problems. The proof of the last theorem is missing.

The following are some errors or drawbacks I found in this paper:

(1) The authors claim that one sample suffices to achieve strong statistical guarantees, but the current proof does not support such strong claim: In line 616 of Supplementary Material, the Frobenius norm of the error is only bounded for sufficiently large n.

This problem may be avoided by considering fixed n and growing dimensions such that the the radius in the definition of $\mathbb{A}$ vanishes. For example, the scaling law considered in line 294 of the main text satisfies the above requirement, and therefore the conclusion is still valid. Only some parts of the proof need to be slightly modified.

(2) In Supplementary Material there are some small errors in proofs: 2.1) Line 607, what is the "boundary" of $\mathbb{A}$? $\mathbb{A}$ is already a sphere-like set, so I see no point talking about the boundary of $\mathbb{A}$.

2.2) Line 661, a typo in the second term of the first equality.

2.3) Line 669, I don't see how the inequality here is applied. What I understood is to simply use sub-multiplicativity of the Frobenius norm and 667 follows.

2.4) Line 699, "primal"-dual witness.

2.5) In the proof of Lemma B.1, the constant  is not explicitly defined. I would suggest to include the definition of  in the statement of the lemma, so as to highlight the fact that the bound in Lemma B.1 depends on || - *||F .

2.6) In the main text the authors state that their proof relies on Talagrand's concentration inequality. Where is the inequality used? Lemma C.2 is not Talagrand's concentration inequality; Gaussian concentration is a classical result that can be simply obtained through, for example, Gaussian log-Sobolev inequality.

(3) Line 303 in the main text, the authors state that their bound is minimax optimal. Minimax with respect to what class?

(4) Line 323, the Hessian := *-1 ? *-1?
Summary: This paper studies the estimation problem when the observations follow tensor normal distribution with separable covariance matrix ($\Sigma^* = \Sigma_1^* \otimes \cdots \otimes \Sigma_K^* $). The authors consider the classical penalized maximum likelihood estimator, which results in a non-convex optimization problem. Quite remarkably, the main result in this paper states that we can simply ignore this non-convexity and do alternating minimization; the so-obtained estimator will achieve strong statistical performance.

Submitted by Assigned_Reviewer_3

Summary: This paper considers tensor graphical models aiming to estimating sparse precision matrices if the overall precision matrix admits a Kronecker product structure with sparse components. It proves various good properties of the alternating minimization algorithm.

Quality: The paper extends some of the early works in the literature by examining tensor data rather than matrix data, noticeably [5, 6, 7, 8].

- The results depend critically on a few assumptions. The first is the irrepresentable condition, which is understood to be quite restricted. The second is (3.4). How can one come up with an initial estimate which lies closes to the truth, when the dimension grows?

- The discussion following Theorem 3.5: The minimax-optimal results have also appeared in [5, 8] when K=2. Thus, the claim that the phenomenon was first discovered

by this paper is not entirely correct. In Remark 3.6, a fair comparison should be made to [8].

- Initial values for the precision matrices: I don't see how and why the suggested initial values (or after iteration) would satisfy (3.4).

- The proposal method in (2.3) is standard. The iterative algorithm is widely used. The theory appears to be standard.

Clarity: The paper is clear.

Originality: The paper makes some contribution to the literature by extending matrix graphical model to tensor graphical model (e.g. in [9]). It is known that the rate in [9] is not optimal in light of [8]. The paper can be best seen as tightening up the results in [9], by using different techniques than those in [8].

Significance: The contribution of the paper is incremental at best.

Summary: A nice attempt was made to solve an interesting problem, but the assumptions are wrong potentially. In some sense the algorithm works in a way similar to [8]. However, how may one find initial values satisfying (3.4) when dimensionality grows?

Submitted by Assigned_Reviewer_4

This paper is clearly written and the symbols are particularly well-defined. It is a pleasure to read through the paper. Some of the theoretical results are direct extensions of the alternating projection algorithm for standard precision matrix estimation. In my opinion, the most helpful quantitative relationship is the equivalence between tensor normal distribution and a multi-dimensional normal distribution (i.e., vec(T)~ N(vec(0); \Sigma_K \otimes...\otimes \Sigma_1)) and more emphasis should be given here to guide the readers (especially those who are already familiar with results for standard precision matrix estimation) through the results for tensor distribution. The shortcoming of this paper is also obvious --- the authors haven't motivated well why this more complex (albeit elegant) model is more suitable for practical problems. I recommend acceptance for its theoretical contribution.
Summary: The authors present an alternating minimization algorithm that obtains the local optimal of the precision matrices of a tensor normal distribution. The problem formulation generalizes the standard estimation problem for sparse precision matrix. I recommend the paper be accepted for its technical contribution.

Submitted by Assigned_Reviewer_5

While the irrepresentability condition (IC) has been largely used in the literature [29,30], it might be useful to include just a few lines regarding the IC for the specific problem of learning Gaussian MRFs, as in [30].

It is important that the authors clarify how Theorem 3.5 depends on \alpha in condition (3.4). A small discussion on how a random initialization satisfies the condition will also be very useful. It might be the case that the assumption is trivial, i.e., \alpha is just a multiplicative factor in the result in Theorem 3.5, but this does not seem clear to me from the paper.

Additionally, the authors are highly encouraged to discuss extensions of their technique on their particular problem and perhaps some other machine learning problems.

There are two statements that somehow seem to imply that learning a simple Gaussian MRF with one sample allows for consistency. Specifically: Line 060: "statistical rate of convergence in Frobenius norm, which is minimax-optimal since this is the best rate one can obtain even when the rest K-1 true precision matrices are known [4]." Line 066: "our alternating minimization algorithm can achieve estimation consistency in Frobenius norm even if we only have access to one tensor sample". It seems that either both statements need qualification or one of the statements is false (e.g., some parts of the proof hold for sufficiently large n, some assumptions bound the Frobenius norm of the Gaussian MRF matrices with dimension-independent constants, etc.)

Few typos (I include things to be added inside [], things to be removed inside {}): Line 177: "is [a] bi-convex problem" Line 181: "by alternatively updat{e}[ing] one precision matrix with [the] other matrices fixed" Line 190: "corresponds to estimating [a] vector-valued Gaussian graphical model"

Summary: Good theoretical results on sample complexity. Good experimental setup.

Author Feedback
Author rebuttal: We thank reviewers for valuable comments. We will revise accordingly in the final version.

Meta_Reviewer

(1) Broader impact.

Our theoretical analysis broadens the usage of the recently developed population-sample analysis techniques for nonconvex statistical problems. See, e.g., Balakrishnan et al. (2014, Statistical guarantees for the EM algorithm: From population to sample-based analysis). These techniques are particularly useful for analyzing statistical optimization problems which have nonconvex sample structure in the worst case but have rather benign population structure in the average case. Besides, with a slight modification, our theoretical analysis can be adapted to establish the estimation consistency for other statistical models with Kronecker product covariance structure, beyond tensor valued graphical models.

(2) Real data.

Data from repeated measurements over time with many features may exhibit temporal dependency as well as dependency across the features. Our tensor graphical model is suitable to detect such dependency structure. Here are two real-world examples: 1. Hoff (2011, Separable covariance arrays via the Tucker product, with applications to multivariate relational data) proposed a tensor normal model for international trade volume data. 2. The high-dimensional AGEMAP microarray data set [3] consists of 8932 gene expression measurements that are recorded on 16 tissue types on 40 mice with varying ages, which forms a four way gene-tissue-mouse-age tensor.

Reviewer_1

(1) Inequality in Line 669.

The inequality was used to bound the second part in line 664. It implies that | \|\hat{M}_1\|_F - \|M_1\|_F | \le \| \hat{M}_1 - M_1 \|_F, and hence line 667 follows.

(2) Concentration inequality.

We actually mean Lemma C.2. We agree with you that it is indeed the concentration for Lipschitz functions of Gaussians. It is used to prove Lemma C.3.

(3) Minimax optimal with respect to what class?

We consider the tensor version of the probability distribution in Cai et al. (2012, Estimating sparse precision matrix: optimal rates of convergence and adaptive estimation).

Reviewer_2

(1) Irrepresentable Condition (IC).

First, IC is not required in our theory for Frobenius norm (Thm 3.5). It is only required for the rates in max norm (Thm 3.9) and spectral norm (Cor 3.10). Second, even for the classical graphical model, IC is standard for achieving the optimal rate in spectral norm (see reference [30]). Third, it is possible to remove IC in proving the rate of convergence in max norm, but additional assumptions on the minimum signal level are necessary (see, e.g., Thm 2 in Loh and Wainwright (2014, Support recovery without incoherence: A case for nonconvex regularization).

(2) Initialization.

Our initialization condition (3.4) trivially holds by setting \alpha=2. Remind that we assume \|\Omega^*_k\|_F=1, k=1,...,K for the identifiability of the tensor normal distribution, see line 153. Thus, any positive definite initialization \Omega_k with unit Frobenius norm satisfies the initialization condition (3.4) with \alpha=2.

(3) Novelty.

Our main contribution is the algorithmic analysis for tensor graphical model, which bridges previous gaps between local and global optima and also establishes the optimal rates. We propose new population-sample analysis techniques, which first establishes one step convergence for the population optimizer (Thm 3.1) and the optimal rate of convergence for the sample optimizer (Thm 3.4)

Reviewer_3

(1) Initialization.

The initialization condition (3.4) trivially holds by setting \alpha=2, since we assume \|\Omega^*_k\|_F=1, k=1,...,K, for the identifiability of the tensor normal distribution, see line 153. In Thm 3.5, \alpha only affects the constant coefficient of the rate, i.e., the constant C_1 in line 675 in the supplement.

(2) Consistency with only one sample.

This phenomenon occurs because the effective sample size for estimating the first way (k=1) precision matrix is n*m_2*m_3*... Even when n=1, the effective sample size still goes to infinity. See the discussion after Thm 3.5, line 293-305.

Reviewer_5

(1) The sparsity of solution.

The estimator of Tlasso is less sparse than that of the P-MLE method. However, a more suitable measure of accuracy is the true positive rate shown in Table 1, which illustrates our advantage.

(2) Initialization.

The insensitivity of the initialization is with respect to the rate of convergence in Thm 3.5. In practice, the initialization affects the constants in the upper bound and may lead to different accuracy. We recommend choosing identity matrix as the initial estimator since it ensures uniformly good performance.

Reviewer_6

(1) Initialization.

The initialization condition (3.4) trivially holds by setting \alpha=2, since we assume \|\Omega^*_k\|_F=1, k=1,...,K, for the identifiability of the tensor normal distribution, see line 153.